# Vision-Language Foundation Models as Effective Robot Imitators

**Xinghang Li**[1,2,†], **Minghuan Liu**[2,3,†], **Hanbo Zhang**[2], **Cunjun Yu**[4], **Jie Xu**[2], **Hongtao Wu**[2],
**Chilam Cheang**[2], **Ya Jing**[2], **Weinan Zhang**[3], **Huaping Liu**[1,✉], **Hang Li**[2], **Tao Kong**[2,✉]
[1]Tsinghua University, [2]ByteDance Research,
[3]Shanghai Jiao Tong University, [4]National University of Singapore
`lixingha23@mails.tsinghua.edu.cn, hpliu@tsinghua.edu.cn,`
`{minghuanliu, wnzhang}@sjtu.edu.cn, kongtao@bytedance.com`

## ABSTRACT

Recent progress in vision language foundation models has shown their ability to understand multimodal data and resolve complicated vision language tasks, including robotics manipulation. We seek a way of making use of existing vision-language models (VLMs) with fine-tuning on robotics data. To this end, we derive a simple and novel vision-language manipulation framework, dubbed *RoboFlamingo*, built upon the open-source VLMs, OpenFlamingo. Unlike prior works, *RoboFlamingo* utilizes pre-trained VLMs for single-step vision-language comprehension, models sequential history information with an explicit policy head, and is slightly fine-tuned by imitation learning only on language-conditioned manipulation datasets. Such a decomposition provides *RoboFlamingo* the flexibility for open-loop control and deployment on low-performance platforms. By surpassing the state-of-the-art performance on the benchmark by a significant margin, we demonstrate that RoboFlamingo presents itself as an effective and competitive alternative for adapting VLMs to robot control. Our extensive experimental results also reveal several interesting conclusions regarding the behavior of different pre-trained VLMs on manipulation tasks. *RoboFlamingo* can be trained or evaluated on a single GPU server, and we believe it has the potential to be a cost-effective and easy-to-use solution for robotics manipulation, empowering everyone with the ability to fine-tune their own robotics policy. Codes and models will be public.

## 1 INTRODUCTION

Recent progress in vision-language foundation models (VLM) has presented their exhilarating ability in modeling and aligning the representation of images and words, and the unlimited potential to resolve a wide range of downstream tasks with multi-modality data, for instance, visual question-answering (Li et al., 2023; Zhou et al., 2022), image captioning (Zeng et al., 2022; Wang et al., 2022; Li et al., 2021), human-agent interactions (Liu et al., 2022b; Oertel et al., 2020; Seaborn et al., 2021). These successes, undeniably, encourage people to imagine a generalist robot equipped with such a vision-language comprehension ability to interact naturally with humans and perform complex manipulation tasks.

Therefore, we aim to explore integrating vision-language foundation models to serve as robot manipulation policies. While there have been some previous studies that incorporated large language models (LLMs) and vision-language models (VLMs) into robot systems as high-level planners (Ahn et al., 2022; Driess et al., 2023), making use of them directly for low-level control still poses challenges. Most VLMs are trained on static image-language pairs, whereas robotics tasks require video comprehension for closed-loop control. Additionally, VLM outputs primarily consist of language tokens, which significantly differ in representation compared to robot actions. A recent work (Brohan et al., 2023), namely Robotics Transformer 2 (RT-2), has demonstrated a possible solution for adapting VLMs to low-level robot control. However, democratizing such an expensive framework for all robotics practitioners proves difficult as it utilizes private models and necessitates

---

†Equal contribution. Works done during the first authors' internship at ByteDance. ✉Corresponding authors.

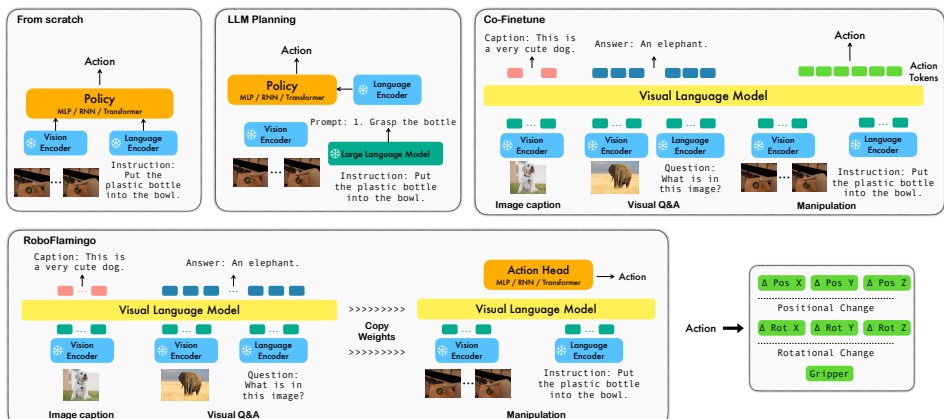

Figure 1: Comparison among *RoboFlamingo* and existing vision-language manipulation solutions.

co-fine-tuning on extensive vision-language data to fully showcase its effectiveness. Consequently, there is an urgent need for robot communities to have a low-cost alternative solution that effectively enables a robot manipulation policy with VLMs.

To this end, we introduce *RoboFlamingo*, a novel vision-language manipulation framework that leverages publicly accessible pre-trained VLMs to effectively construct manipulation policies for robotics. Specifically, *RoboFlamingo* is grounded upon the open-source VLM, OpenFlamingo (Awadalla et al., 2023), and resolves the challenge by decoupling visual-language understanding and decision-making. Unlike previous works, *RoboFlamingo* takes advantage of pre-trained VLMs mainly for understanding vision observations and language instructions at every decision step, models the historical features with an explicit policy head, and is fine-tuned solely on language-conditioned manipulation datasets using imitation learning. With such a decomposition, we only need to combine a small amount of robotics demonstration to adapt the model to downstream manipulation tasks, and *RoboFlamingo* also offers flexibility for open-loop control and deployment on low-performance platforms. Moreover, benefiting from the pre-training on extensive vision-language tasks, *RoboFlamingo* achieves state-of-the-art performance with a large margin over previous works, and generalizes well to zero-shot settings and environments. It is worth noting that *RoboFlamingo* can be trained or evaluated on a single GPU server. As a result, we believe *RoboFlamingo* can be a cost-effective yet high-performance solution for robot manipulation, empowering everyone with the ability to fine-tune their own robots with VLMs.

Through extensive experiments, we demonstrate that *RoboFlamingo* outperforms existing methods by a clear margin. Specifically, we evaluate its performance using the Composing Actions from Language and Vision benchmark (CALVIN) (Mees et al., 2022b), a widely-recognized simulation benchmark for long-horizon language-conditioned tasks. Our findings indicate that *RoboFlamingo* is an effective and competitive alternative for adapting VLMs to robot control, achieving a performance improvement that is two times greater compared to the previous state-of-the-art method. Our comprehensive results also yield valuable insights into the use of pre-trained VLMs for robot manipulation tasks, offering potential directions for further research and development.

## 2 RELATED WORK

Language can be the most intuitive and pivotal interface for human-robot interaction, enabling non-expert humans to seamlessly convey their instructions to robots for achieving diverse tasks. Consequently, the realm of language-conditioned multi-task manipulation has garnered substantial attention in recent years. Intuitively, such tasks require robots to have a good understanding of not only the visual captures of the outside world, but also the instructions represented by words. With the strong representation ability of pre-trained vision and language models, a lot of previous works have incorporated pre-trained models into the learning framework. Among them, we roughly classify them into the following three categories, which is also illustratively compared in Fig. 1.

**Fine-tuning.** While some early works such as Jang et al. (2022); Lynch & Sermanet (2020) trained a vision encoder and a language encoder to learn representations for the input language and vision data from manipulation tasks, some recent work directly takes pre-trained models to obtain great representations, then trains the policy model beyond them from scratch or fine-tuning the whole model. For instance, Jiang et al. (2023) utilizes a pre-trained T5 (Raffel et al., 2020) model to encode the multi-modal prompts, and learn the actions by fine-tuning the T5 model and additionally training an object encoder and attention layers. HULC (Mees et al., 2022a) utilizes the vision encoder of Lynch & Sermanet (2020) trained on the CALVIN dataset (Mees et al., 2022b) and some pre-trained language encoder models such as sentence transformer (Reimers & Gurevych, 2019), and their HULC++ (Mees et al., 2023) also fine-tunes these encoders. Besides, Brohan et al. (2022) proposed RT-1, i.e., robotics transformers, a 35M vision-language-action model (VLA) which tokenizes the action and aligns the vision, language, and action in the token space and is trained on a large amount of real-world manipulation dataset, using the Universal Sentence Encoder (Cer et al., 2018) to obtain the language embedding and the pre-trained EfficientNet-B3 (Tan & Le, 2019) as the vision tokenizer.

**LLM planning.** Some approaches have exploited large language models (LLMs) as a powerful zero-shot planner, e.g., SayCan Ahn et al. (2022), to generate step-by-step pre-defined plans with human-interactive prompts on given tasks, subsequently instructing different pre-trained low-level skill policies to execute those plans and finish multiple tasks. Compared to other works, the controlling policies do not require any ability to understand instructions, but rely on the pre-trained frozen LLM to select necessary skills.

**Co-Fine-Tuning.** Driess et al. (2023) proposed 540B PaLM-E model, showing a different way of utilizing the pre-trained vision and language model. Specifically, they choose different pre-trained models to encode the input scene, and the PaLM (Chowdhery et al., 2022) model as the base model, train the model to generate pre-defined multi-step plans described by language by co-fine-tuning the whole VLM end-to-end using both mobile manipulation question-answering data and auxiliary vision-language training data such as image captioning and visual question answering data collected from the web. Similar to SayCan (Ahn et al., 2022), they require low-level control policies to execute the generated plans. Motivated by PaLM-E, Brohan et al. (2023) further introduced RT-2, which is based on RT-1 but is adapted to use large vision-language backbones like PaLI-X (Chen et al., 2023) and PaLM-E (Driess et al., 2023), training the policy utilizing both robot manipulation data and web data. Their method reveals that VLMs have the potential to be adapted into robot manipulation, yet their key co-fine-tuning training strategy requires a large amount of both web-scale data vision-language data and low-level robot actions. Additionally, the VLMs and the data they use are private, making it hard for every robotics practitioner to play on such a solution for their own.

Although these previous models somehow bridge the gap between vision and language on robot manipulation tasks, they either reply on low-level skill policies, like SayCan and PaLM-E; or train a whole large model, such as RT-1; or require a huge amount of vision-language data and computational resources to ensure the model learns the manipulation policy without forgetting the great alignment of vision and language. Compared with these works, our proposed *RoboFlamingo* is a simple and intuitive solution to easily adapt existing VLMs (OpenFlamingo (Alayrac et al., 2022; Awadalla et al., 2023) used in this paper), only requiring fine-tuning on a small number of manipulation demonstrations. We hope *RoboFlamingo* provides a different perspective on fully leveraging the ability of VLMs, while requiring less data collection costs and computing consumption to make it an open and easy-to-use solution for everyone.

## 3 BACKGROUND

**Robot manipulation.** In this paper, we mainly consider robot manipulation tasks, where the agent (robot) does not have access to the ground-truth state of the environment, but visual observations from different cameras and its own proprioception states. As for the action space, it often includes the relative target pose and open/closed state of the gripper. For instance, in the testbed of CALVIN (Mees et al., 2022b), the observations consist of simulated camera captures from two different views, and the action is a 7-DoF control of a Franka Emika Panda robot arm with a parallel gripper, and the instructions are reaching goals, i.e., the after-the-fact descriptions.

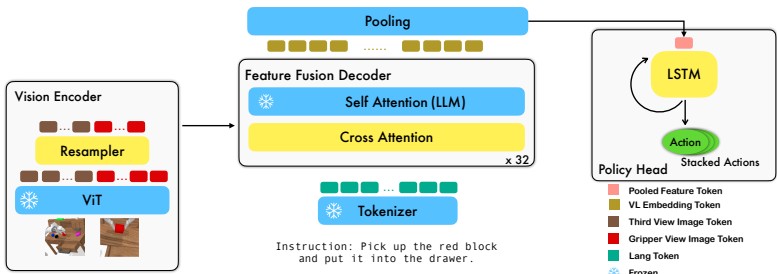

Figure 2: The illustration of the proposed *RoboFlamingo* framework. The Flamingo backbone models single-step observations, and the temporal features are modeled by the policy head.

**Imitation learning.** Imitation learning (Pomerleau, 1988; Zhang et al., 2018; Liu et al., 2020; Jang et al., 2022) allows the agent to mimic the manipulation plans from instruction-labeled expert play data $\mathcal{D} = \{(\tau, l)_i\}_{i=0}^{D}$, where $D$ is the number of trajectories, $l$ is the language instruction, and $\tau = \{(o_t, a_t)\}$ contains preceding states and actions to reach the goal described by the given instruction. The learning objective can be simply concluded as a maximum likelihood goal-conditioned imitation objective to learn the policy $\pi_\theta$:

$$\ell = \mathbb{E}_{(\tau, l)_i \sim \mathcal{D}} \left[ \sum_{t=0}^{|\tau|} \log \pi_\theta(a_t | o_t, l) \right]. \tag{1}$$

## 4 ROBOFLAMINGO

*RoboFlamingo*, a generalized robotics agent, excels in resolving language-conditioned manipulation tasks. The key idea is to draw help from pre-trained vision-language models (VLMs) and adapt them to manipulation policies, acquiring the ability of object grounding, language comprehension, vision-language alignment, and long-horizon planning. Particularly, *RoboFlamingo* looks into one of the popular VLMs, Flamingo (Alayrac et al., 2022), and takes its open-source model Open-Flamingo (Awadalla et al., 2023) as the backbone. The overview of *RoboFlamingo* is shown in Fig. 2. To adapt large-scale vision-language models to robotic manipulation, *RoboFlamingo* simply adds a policy head for end-to-end finetuning. It addresses three main challenges: 1) it adapts vision-language models with static image inputs to video observations; 2) it generates robot control signals instead of text-only outputs; 3) it requires a limited amount of downstream robotic manipulation data to achieve high performance and generality with billions of trainable parameters. We will elaborate on the design of *RoboFlamingo* in this section.

### 4.1 LANGUAGE-CONDITIONED ROBOT CONTROL

The problem of language-conditioned robot control can be modeled as a goal-conditioned partially observable Markov decision process (GC-POMDP) (Liu et al., 2022a): $\mathcal{M} = \langle \mathcal{S}, \mathcal{O}, \mathcal{A}, \mathcal{T}, \rho_0, \mathcal{L}, \phi, f \rangle$, where $\mathcal{S}$ and $\mathcal{O}$ are the set of states and observations separately, $\mathcal{A}$ is the action space, $\mathcal{T} : \mathcal{S} \times \mathcal{A} \rightarrow \mathcal{S}$ is the environment dynamics function, $\rho_0 : \mathcal{S} \rightarrow [0, 1]$ is the initial state distribution, $\phi(s)$ indicate if the task is successful, and $f(o|s) : \mathcal{S} \rightarrow \mathcal{O}$ is the observation function. Specifically, for each controlling episode, the robot is given a goal, represented by a length-$M$ free-form language instruction $l \in \mathcal{L}$ at every time step $t$, and the observations $o_t$ are typically two images $I_t$, $G_t$ from a third-perspective camera and a gripper camera. The controlling policy can be modeled as a goal-conditioned policy $\pi(a|o, l) : \mathcal{S} \times \mathcal{L} \rightarrow \mathcal{A}$ and the action $a$ is typically the desired relative position and pose of the gripper, along with its open/close status.

In our *RoboFlamingo*, the policy $\pi_\theta(a|o, l)$ is parameterized by $\theta$. It consists of a backbone based on Flamingo $f_\theta$ and a policy head $p_\theta$. The backbone takes visual observations and language-represented goals as the input and provides a latent fused representation at each time step for the policy head: $X_t = f_\theta(o_t, l)$. Then the policy head further predicts the action to fulfill the specified goal for the robot: $a_t = p_\theta(X_t, h_{t-1})$, where $h_{t-1}$ is the hidden state from the last step that encodes the history information for decision-making. We will introduce each module in detail in the following sections.

## 4.2 THE FLAMINGO BACKBONE

We adopt the Flamingo backbone $f_\theta$ for understanding the vision and language inputs at every decision step. Overall, Flamingo encodes the vision observations to the latent tokens by a vision encoder; and then fuses them with language goals through the feature fusion decoder. We explain these parts in detail below.

### 4.2.1 VISION ENCODER

The vision encoder consists of a vision transformer (ViT) (Yuan et al., 2021) and a perceiver resampler (Alayrac et al., 2022). At every time step $t$, the two-view camera images $I_t$, $G_t$ are encoded to $\hat{X}_t$, consisting of a visual token sequence, through the ViT module:

$$\hat{X}_t^v = \text{ViT}(I_t, G_t), \tag{2}$$

where $\hat{X}_t^v = (\hat{x}_{t1}^v, \cdots, \hat{x}_{tN}^v)$ represents the visual token sequence at $t$, $N$ represents the token number of the encoded output. After encoding, *RoboFlamingo* utilizes a perceiver resampler to compress the number of visual tokens from $N$ to $N_r$. In detail, the resampler maintains a set of learnable parameters and utilizes the attention mechanism to reduce the number of token sequences to $K$. Formally, the resampler is formulated as:

$$K_R = \hat{X}_t^v W_K^R, \;\; V_R = \hat{X}_t^v W_V^R, \;\; X_t^v = \text{softmax}(\frac{Q_R K_R^T}{\sqrt{d}})V_R, \tag{3}$$

where $Q_R \in \mathbb{R}^{N_r \times d}$ corresponds to the learnable parameters of the resampler and serves as the query vector, $d$ is the hidden dimension size, $W_K^R, W_V^R \in \mathbb{R}^{d_v \times d}$ represents the linear transformation matrix of key and value, $d_v$ is the feature dimension of the visual token, $K_R$ and $V_R$ are the transformed key and value vector of vision input $V$.

### 4.2.2 FEATURE FUSION DECODER

The compressed visual tokens output from the resampler $X_t^v \in \mathbb{R}^{N_r \times d}$ are further passed to the feature fusion decoder, which is designed to generate the vision-language joint embedding by fusing the language instruction with the encoded vision feature $X_t^v$. In *RoboFlamingo*, we utilize the pre-trained decoder from OpenFlamingo (Awadalla et al., 2023) and fine-tune the decoder module following the way as in Awadalla et al. (2023). Specifically, the decoder consists of $L$ layers, each of which involves a transformer decoder layer and a cross-attention layer. The transformer layers are directly copied from a pre-trained language model (such as LLaMA (Touvron et al., 2023), GPT-Neox (Black et al., 2022) and MPT (Team et al., 2023)) and are frozen during the whole training process; the cross-attention layer takes the language token as query, and the encoded visual token as key and value, which is fine-tuned by imitation learning objectives on manipulation data (see following sub-sections). Formally, if we denote $x_i \in \mathbb{R}^d$ the $i-$th embedded token of the instruction, $M$ the instruction length, and $X \in \mathbb{R}^{M \times d}$ is the embedded matrix of the instruction, then the embedded natural language instruction should be $X = (x_1, x_2, \cdots, x_M)$ and output $X_t^{l+1}$ of the $l$-th decoder layer given the input $X_t^l$ is computed by:

$$\begin{aligned} \hat{X}_t^l &= \text{Tanh}(\alpha) \cdot \text{MLP}(A(X_t^l W_Q^C, X_t^v W_K^C, X_t^v W_V^C)) + X_t^l, \\ X_t^{l+1} &= \text{MLP}(A(\hat{X}_t^l W_Q^S, \hat{X}_t^l W_K^S, \hat{X}_t^l W_V^S)) + \hat{X}_t^l, \end{aligned} \tag{4}$$

where $X_t^1 = X$, $\hat{X}_t^l$ corresponds to the output of the gated cross-attention layer at time instant $t$, $W_Q^C, W_K^C, W_V^C \in \mathbb{R}^{d \times d}$ represents the learnable parameters of the cross-attention layer. $\alpha \in \mathbb{R}$ is a learnable gate parameter to control the mixing weights for stability. $W_Q^S, W_K^S, W_V^S \in \mathbb{R}^{d \times d}$ represents the parameters of the self-attention layer and MLP represents a multi-layer perceptron network. With the deep interaction of the vision and language token, we expect the output $X_t = X_t^L = \{x_{t,1}^L, x_{t,2}^L, \cdots, x_{t,M}^L\}$ at time step $t$ to be an informative vision-language joint embedding for robot manipulation.

## 4.3 POLICY HEAD

The output $X_t^L$ from the feature fusion decoder is trained as the representation of the vision observation and language instruction, which will be further translated into low-level control signals. To

achieve this, we simply adopt an additional policy head $p_\theta$ to predict the action, e.g., the 7 DoF end-effector pose and gripper status. We test various strategies to model the historical observation sequences and behave as the policy head, e.g., a long short-term memory (LSTM) (Hochreiter & Schmidhuber, 1997) network with an MLP for the final prediction; a decoder-only transformer (Brown et al., 2020) similarly with an MLP; or a single MLP that only models single-step information (see Section 5 for more details). Taking the LSTM version as an example, with the vision-language joint embedding sequence $X_t^L$, we obtain an aggregated embedding through a max-pooling operation over the token dimension and predict the action as:

$$\tilde{X}_t = \text{MaxPooling}(X_t); h_t = \text{LSTM}(\tilde{X}_t, h_{t-1}); a_t^{pose}, a_t^{gripper} = \text{MLP}(h_t) , \qquad (5)$$

where $h_t$ represents the hidden state at $t$, and $a_t^{pose}, a_t^{gripper}$ are the predicted end-effector pose and gripper status.

### 4.4 TRAINING OBJECTIVE

We utilize maximum likelihood imitation learning objectives to fine-tune the proposed pre-trained backbone and the policy head. Concretely, the desired relative pose is optimized via regression loss (we use mean squared error (MSE) loss) and the gripper status uses classification loss (we use binary cross-entropy (BCE) loss):

$$\ell = \sum_t \text{MSE}(a_t^{pose}, \hat{a}_t^{pose}) + \lambda_{gripper} \text{BCE}(a_t^{gripper}, \hat{a}_t^{gripper}), \qquad (6)$$

where $\hat{a}_t^{pose}, \hat{a}_t^{gripper}$ is the demonstration for end effector pose and gripper status at timestep $t$, $\lambda_{gripper}$ corresponds to the weight of gripper loss.

In the training procedure, we follow the fine-tuning paradigm of OpenFlamingo by only training the parameters of the resampler, the gated cross-attention module of each decoder layer, and the policy head while freezing all other parameters.

## 5 EXPERIMENTS

We conduct extensive experiments to examine the proposed *RoboFlamingo* solution, and answer how pre-trained VL models (VLMs) benefit language-conditioned robotic manipulation. In short, we investigate *RoboFlamingo* from the following perspectives:

1. **Effectiveness.** We wonder the imitation learning performance of *RoboFlamingo* by training it on the given demonstration data.

2. **Zero-shot Generalization.** We focus on generalization on unseen tasks. In other words, we study how the model will behave given unseen vision contexts like different objects, even with unseen instructions.

3. **Ablation Studies.** We further explore the essential factors that matter in adapting VLMs to robot control policy in the framework of *RoboFlamingo*.

### 5.1 BENCHMARK AND BASELINES

We choose CALVIN (Mees et al., 2022b), an open-source simulated benchmark to learn long-horizon language-conditioned tasks, as our testbed, and the corresponding datasets as our imitation learning demonstration data. CALVIN encompasses a total of 34 distinct tasks and evaluates 1000 unique instruction chains for sequential tasks. In each experiment, the robot is required to successfully complete sequences of up to five language instructions consecutively. The policy for each consecutive task is dependent on a goal instruction, and the agent advances to the subsequent goal only if it successfully accomplishes the current task. The dataset contains four splits for environments A, B, C, and D. Each consists of 6 hours of human-teleoperated recording data (more than 2 million steps) that might contain sub-optimal behavior, and only 1% of that data is annotated with language instructions (∼24 thousand steps). See Fig. 4 in Appendix A.1 for a more detailed description and visualized examples of the benchmark.

Table 1: The imitation performance on various settings, all results are reported using the best-behaved model checkpoints. *Full* and *Lang* denote if the model is trained using unpaired vision data (i.e., vision data without language pairs); *Freeze-emb* refers to freezing the embedding layer of the fusion decoder; *Enriched* denote using GPT-4 enriched instructions. The gray rows denote numerical results evaluated by our re-trained model. We re-implement RT-1 and take the original code of HULC provided by Mees et al. (2022a). All other results are reported by Mees et al. (2022a).

| Method | Training Data | Test Split | Task Completed in a Sequence (Success Rate) | | | | | |
|---|---|---|---|---|---|---|---|---|
| | | | 1 | 2 | 3 | 4 | 5 | Avg Len |
| MCIL | ABCD (Full) | D | 0.373 | 0.027 | 0.002 | 0.000 | 0.000 | 0.40 |
| HULC | ABCD (Full) | D | 0.889 | 0.733 | 0.587 | 0.475 | 0.383 | 3.06 |
| HULC | ABCD (Lang) | D | 0.892 | 0.701 | 0.548 | 0.420 | 0.335 | 2.90 |
| RT-1 | ABCD (Lang) | D | 0.844 | 0.617 | 0.438 | 0.323 | 0.227 | 2.45 |
| *RoboFlamingo* (Ours) | ABCD (Lang) | D | **0.964** | **0.896** | **0.824** | **0.740** | **0.66** | **4.09** |
| MCIL | ABC (Full) | D | 0.304 | 0.013 | 0.002 | 0.000 | 0.000 | 0.31 |
| HULC | ABC (Full) | D | 0.418 | 0.165 | 0.057 | 0.019 | 0.011 | 0.67 |
| RT-1 | ABC (Lang) | D | 0.533 | 0.222 | 0.094 | 0.038 | 0.013 | 0.90 |
| *RoboFlamingo* (Ours) | ABC (Lang) | D | **0.824** | **0.619** | **0.466** | **0.331** | **0.235** | **2.48** |
| HULC | ABCD (Full) | D (Enriched) | 0.715 | 0.470 | 0.308 | 0.199 | 0.130 | 1.82 |
| RT-1 | ABCD (Lang) | D (Enriched) | 0.494 | 0.222 | 0.086 | 0.036 | 0.017 | 0.86 |
| Ours | ABCD (Lang) | D (Enriched) | 0.720 | 0.480 | 0.299 | 0.211 | 0.144 | 1.85 |
| Ours (freeze-emb) | ABCD (Lang) | D (Enriched) | **0.737** | **0.530** | **0.385** | **0.275** | **0.192** | **2.12** |

We compare a set of well-performed baselines in CALVIN: (1) MCIL (Lynch & Sermanet, 2020): a scalable framework combining multitask imitation with free-form text conditioning, which learns language-conditioned visuomotor policies, and is capable of following multiple human instructions over a long horizon in a dynamically accurate 3D tabletop setting. (2) HULC (Mees et al., 2022a): a hierarchical method that combines different observation and action spaces, auxiliary losses, and latent representations, which achieved the SoTA performance on CALVIN. (3) RT-1 (Brohan et al., 2022): robotics transformer, which directly predicts the controlling actions by action tokens, as well as vision and language inputs. RT-2 (Brohan et al., 2023) is not experimentally compared since we have no access to their code, data, and model weights.

## 5.2    IMITATION PERFORMANCE

We train *RoboFlamingo* (with the M-3B-IFT backbone) using demonstrations only with language annotation from all 4 splits (A, B, C, and D), and evaluate the imitation performance on episodes sampled on split D ($ABCD \rightarrow D$).The performance comparison is shown in Tab. 1. *RoboFlamingo* outperforms all baseline methods over all metrics by a large margin, even for those methods that are trained on the full set of data. This demonstrates the effectiveness of *RoboFlamingo* as the solution for robotics manipulation, enabling VLMs to become effective robot imitators.

In addition, the success rate of the subsequent tasks can be regarded as a notion of the generalizability of the manipulation policies, since the initial state of a subsequent task highly relies on the ending state of its former task. The later a task is arranged in the task sequence, the more diverse its initial state is, which will need more powerful visual-language alignment abilities to successfully complete the task. Among all methods, *RoboFlamingo* achieves the highest success rate over the latter tasks. This demonstrates that *RoboFlamingo* is able to utilize the visual-language grounding ability of pre-trained VLMs. In the appendix, we further include the results of *RoboFlamingo* co-trained with COCO and VQA data (Appendix B.1) and compare with recent robotics representation works (Appendix B.2). Appendix B.1 also reveals how the original VL abilities change after fine-tuning.

## 5.3    ZERO-SHOT GENERALIZATION

To assess the zero-shot generalization ability, we evaluate *RoboFlamingo* in two aspects: vision and language. For vision generalization, we train models on splits A, B, and C and test on split D, which presents a different vision context. Our method significantly outperforms baselines in this vision generalization scenario ($ABC \rightarrow D$), as shown in Tab. 1. Regarding language generalization, we enrich the language setting by generating 50 synonymous instructions for each task using GPT-4 (Achiam et al., 2023). We then randomly sample instructions during evaluation. Our method exhibits superior performance compared to all baselines in this language generalization setting.

Table 2: Variants of VLMs tested. *Pre-train* denotes the original performance of VLM on the pre-training VL dataset, *Best Avg. Len.* denotes the best performance of the average success length of VLMs within 5 epochs, and *Mean Avg. Len.* denotes the mean performance of the average success length of VLMs of the last 3 epochs on CALVIN.

| Backbone Name | LLM Arch | Total Param | LLM Param | Trainable Param | Instr. Tuning | Pre-trained (Public, 4-shot) | | Avg. Len. | |
|---|---|---|---|---|---|---|---|---|---|
| | | | | | | COCO (CIDEr) | VQAv2 (Acc) | Best | Mean |
| M-3B | MPT | 3B | 1B | 1B | ✗ | 77.3 | 45.8 | 3.94 | 3.81 |
| M-3B-IFT | | | | | ✓ | 82.7 | 45.7 | **4.09** | **4.02** |
| G-4B | GPT-Neox | 4B | 3B | 1B | ✗ | 81.8 | 49.0 | 3.67 | 3.53 |
| G-4B-IFT | | | | | ✓ | 85.8 | 49.0 | 3.79 | 3.72 |
| L-9B | LLaMA | 9B | 7B | 1B | ✗ | 74.3 | 44.0 | 2.79 | 2.71 |
| M-9B | MPT | | | | ✗ | **89.0** | **54.8** | 3.97 | 3.87 |

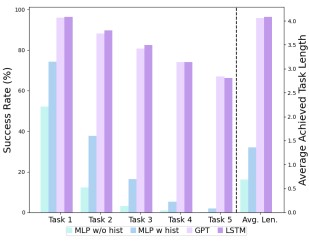

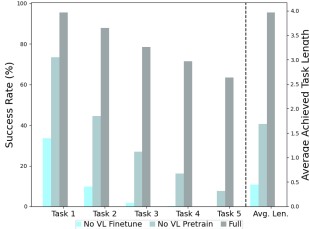

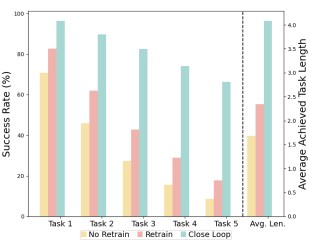

| (a) Various policy formulation. | (b) Different training paradigms. | (c) Open loop control. |

Figure 3: Ablation studies on the $ABCD \rightarrow D$ setting.

Note that the success rate of *RoboFlamingo* on subsequent tasks dropped more than HULC does. This may be due to our approach directly using word tokens as input during training, which can result in larger variations for synonymous sentences compared to HULC using a frozen sentence model for embedding instructions. To address this, we freeze the embedding layer of the feature fusion decoder in our method, leading to improved generalization and reduced performance drop.

## 5.4 ABLATION STUDIES

In this section, we conduct ablation studies for *RoboFlamingo* to answer the following questions:

    1) How does *RoboFlamingo* perform with different policy heads/formulations?

    2) Does vision-language (VL) pre-training improve downstream robotic tasks?

    3) How do critical factors in VL pre-training affect robotic tasks?

### 5.4.1 HOW DOES *RoboFlamingo* PERFORM WITH DIFFERENT POLICY FORMULATIONS?

We test *RoboFlamingo* with different policy heads/formulations. In particular, we compare 4 different implementations: (a) $MLP\ w/o\ hist$ takes only the current observation as input to predict actions, which ignores the observation history. (b) $MLP\ w\ hist$ takes the history frames into the vision encoder with position embedding, and encodes the history information through the cross-attention layers in the feature fusion decoder. (c) $GPT$ and (d) $LSTM$ both utilize the VLM backbone to process single-frame observations and integrate the history with the policy head. $GPT$ explicitly takes the visual history as input to predict the next action. $LSTM$ implicitly maintains a hidden state to encode memory and predict the action. See Appendix C.1 for detailed illustration. We compare their best performance on the $ABCD \rightarrow D$ setting in Fig. 3 (a). $MLP\ w/o\ hist$ performs the worst, indicating the importance of the history information in the manipulation task. $MLP\ w\ hist$ performs better than $MLP\ w/o\ hist$, but is still much worse than $GPT$ and $LSTM$. We hypothesize that this may stem from the fact that the VLM (OpenFlamingo) has only seen image-text pairs during pre-training and cannot process consequent frames effectively. Further, the performance of $GPT$ and $LSTM$ are similar, we choose $LSTM$ as the default choice due to its simplicity.

### 5.4.2 DOES VL PRE-TRAINING IMPROVE DOWNSTREAM ROBOTIC TASKS?

To verify the necessity of VL pre-training, we train the same model without loading the pre-trained parameters of the cross-attention layers and the resampler trained by OpenFlamingo models (denoted

Table 3: The performance on 10% language annotated data on $ABCD \rightarrow D$ setting. All variants are trained and evaluated for the same training epochs.

| Method | Task Completed in a Sequence (Success Rate) | | | | | |
|--------|-------|-------|-------|-------|-------|---------|
|        | 1     | 2     | 3     | 4     | 5     | Avg Len |
| M-3B       | 0.047 | 0.003 | 0.000 | 0.000 | 0.000 | 0.05 |
| M-3B-IFT   | 0.120 | 0.007 | 0.000 | 0.000 | 0.000 | 0.13 |
| G-4B       | 0.420 | 0.054 | 0.003 | 0.000 | 0.000 | 0.48 |
| G-4B-IFT   | 0.448 | 0.084 | 0.014 | 0.003 | 0.001 | 0.55 |
| M-9B       | 0.547 | 0.190 | 0.067 | 0.020 | 0.003 | 0.83 |

as *No VL Pre-train*). Besides, we also conduct an ablation study to freeze the pre-trained VLM and only train the policy head (denoted as *No VL Finetune*). As shown in Fig. 3 (b), we can see that vision-language pre-training crucially improves the downstream robotic manipulation by a large margin. Besides, tuning on the VL model itself on robotic tasks is indispensable due to the limited capacity of the policy head.

### 5.4.3 HOW DO CRITICAL FACTORS IN VL PRE-TRAINING AFFECT ROBOTIC TASKS?

**Model size.** A larger model usually results in better VL performance. Yet, with full training data in CALVIN, we find that the smaller model is competitive with the larger model (see the comparison in Tab. 2 and Appendix B.4). To further validate the impact of model size on downstream robotic tasks, we train different variants with 10% of language annotated data in CALVIN, which is only 0.1% of the full data. From Tab. 3 we can observe that with limited training data, the performance of VLMs is highly related to the model size. The larger model achieves much higher performance, indicating that a larger VLM can be more data-efficient.

**Instruction fine-tuning.** Instruction-Finetuning is a specialized technique that utilizes a further pre-training enhancement on the LLM with the IFT dataset (Conover et al., 2023; Peng et al., 2023), which provides a rich repertoire of instruction-following behaviors that inform its capabilities in language-conditioned tasks. We find that LLMs with such a training stage can improve the performance of the policy in both seen and unseen scenarios, revealed by the performance improvements of M-3B-IFT against M-3B, and G-4B-IFT against G-4B shown in Tab. 2.

### 5.5 FLEXIBILITY OF DEPLOYMENT

Since our *RoboFlamingo* adopts a structure that separates the perception and policy module and leaves the main computation on the perception module, we could perform open loop control to accelerate the inference of *RoboFlamingo*. Instead of taking only the next action to execute and performing VLM inference every time for new observations to predict future actions, open-loop control can be achieved by predicting an action sequence (stacked actions) with only one inference given the current observation, therefore alleviating the delay and the test-time computing requirement. However, as indicated in Fig. 3 (c), directly implementing open loop control without re-training may lead to deteriorated performance, retraining the model with jump step demonstration could alleviate the performance drop.

## 6 CONCLUSION AND FUTURE WORK

This paper explores the potential of pre-trained vision-language models in advancing language-conditioned robotic manipulation. Our proposed *RoboFlamingo*, based on the pre-trained Open-Flamingo model, showcases state-of-the-art performance on a benchmark dataset. Moreover, our experimental findings highlight the benefits of pre-trained models in terms of data efficiency and zero-shot generalization ability. This research contributes to the ongoing efforts to develop intelligent robotic systems that can seamlessly understand and respond to human language instructions, paving the way for more intuitive and efficient human-robot collaboration. Due to the lack of real-robot data, this paper does not deploy on real-world robotics. To our delight, recent progress on large-scale real robotics data (Padalkar et al., 2023) has shown the potential of fine-tuning large VLMs for real robots, and the most exciting future work is to see how *RoboFlamingo* will behave in real-world tasks combined with such amount of data.

ACKNOWLEDGEMENTS

This work was supported by the National Natural Science Foundation of China under Grant 62025304. The Shanghai Jiao Tong University team is partially supported by National Key R&D Program of China (2022ZD0114804), Shanghai Municipal Science and Technology Major Project (2021SHZDZX0102) and National Natural Science Foundation of China (62322603, 62076161). The author Minghuan Liu is also supported by the ByteDance Scholarship and Wu Wen Jun Honorary Doctoral Scholarship.

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

## A  ENVIRONMENTAL SETUPS

### A.1  THE CALVIN BENCHMARK

CALVIN (Mees et al., 2022b) is an open-source simulated benchmark for evaluating long-horizon language-conditioned tasks.

As shown in Fig. 4, CALVIN includes four different environments A, B, C, and D, each of which consists of 6 hours of human-teleoperated recording data (more than 2 million trajectories) that might contain sub-optimal behavior, and only 1% of that data is annotated with language instructions (around 24 thousand trajectories). Each split is settled with different settings of objects and environments, aiming to validate the performance, robustness, and generality of policies trained with different data combinations.

This benchmark requires a 7-DOF Franka Emika Panda robot arm with a parallel gripper, utilizing onboard sensors and images from two camera views to successfully complete sequences of up to five language instructions consecutively. This setup further challenges the robot's ability to transition between various goals. CALVIN encompasses a total of 34 distinct tasks and evaluates 1000 unique instruction chains for sequences. The robot is reset to a neutral position after each sequence to prevent any policy bias resulting from its initial pose. This neutral initialization eliminates any correlation between the initial state and the task, compelling the agent to rely solely on language cues to comprehend and solve the given task. The policy for each consecutive task is dependent on the instruction of the current goal, and the agent advances to the subsequent goal only if it successfully accomplishes the current task.

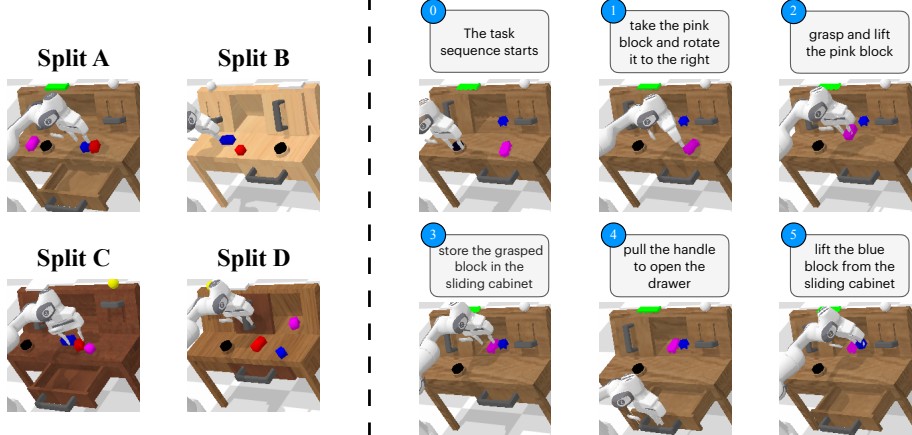

Figure 4: The visualization of the four splits (left) and a full task sequence demonstration in CALVIN (right). The ID in the blue circle represents the end of which task among the five tasks. When the task is finished, the instructions for the next task will be given. For instance, the second image in the first row of the right side denotes the end of task 1, and at the next step the instruction shown in the blue circle "2" will be given. Blue circle 0 is not an instruction.

### A.2  EXAMPLES OF ENRICHED INSTRUCTIONS

Table 4: Examples of original and enriched instructions in the CALVIN benchmark.

| Task Type | rotate red block right | push pink block | move slider left | open drawer | lift blue block slider |
|---|---|---|---|---|---|
| CALVIN Instruction | Take the red block and rotate it to the right | Go push the pink block left | Push the sliding door to the left side | Pull the handle to open the drawer | Lift the blue block from the sliding |
| Enriched Instruction | Rotate the red item in a clockwise direction | Shift the pink block to the left | Push the gliding doorway to the left | Grasp the handle firmly and pull to dislodge the drawer | Carefully hist the blue marker out of the mobile drawer |
| | Give a rightward spin to the red block | Roll the pink cube on the left | Use your arm to slide the door towards the left | Grip the handle exert force to unfold the drawer | Lift upward the blue block from the sliding closet |
| | Change the position of the red block to the right | Dislocate the pink cube to your left | Guide sliding passageway to the left | Tug the handle to make the drawer slide out | Grasp and lift the blue box from the rolling drawer |

To validate the performance of the policies over diversified language expressions, we utilize GPT4 to augment the language instruction in CALVIN. We showcase the enriched language instructions in Tab. 4. We can see that the enriched instructions do have the same meaning as the original one, yet they are organized with different words. As shown in Table 1, *RoboFlamingo* can still achieve better performance compared to HULC.

### A.3 COMPUTING RESOURCE

All experiments involved in this paper are conducted on a single GPU server with 8 NVIDIA Tesla A100 GPUs, and the default batch size is 6 on each GPU. The MPT-3B model takes 13 hours of training per epoch and achieves the best performance at the 3rd epoch, while the MPT-9B model also takes 26 hours of training per epoch and achieves the best performance at the 4rd epoch.

## B EXTENDED EXPERIMENTAL RESULTS

### B.1 CO-TRAINING

From Tab. 1, the *Enriched* setting, we have noticed some evidence that the model may lose some foundation capabilities as the performance loss, which indicates that there is over-fitting during the fine-tuning. To further understand the phenomenon, we conduct further experiments by testing the fine-tuned RoboFlamingo model (the M-3B-IFT variant) on the COCO image caption and VQAv2, which verify our conjecture (see Tab. 6). To prevent such problems, we choose to co-train RoboFlamingo (the M-3B-IFT variant) with VQA and COCO datasets during fine-tuning on the robotics dataset. We test the co-train model on CALVIN, and the COCO image caption, VQAv2 tasks as well, as shown in Tab. 6 and Tab. 5. This provides a solution for fine-tuning VLMs to robotics models while preserving the ability on vision-language tasks, even though it may slightly deteriorate the performance on robotic tasks. In our implementation, we ensure that the model equally incorporates batches of VL and robot data in each epoch. From Fig. 5 and Fig. 5 we could observe a similar performance curve of the Co-trained and Fine-tune version of our model, while Co-trained model achieves higher performance in the early epochs and Fine-tune model ends up higher in the later epochs. One interesting observation is that under the *Enriched* setting, the performance of the co-trained model also drops, this may indicate the difference between understanding different sentences and aligning vision-language representations (as the pre-trained tasks do).

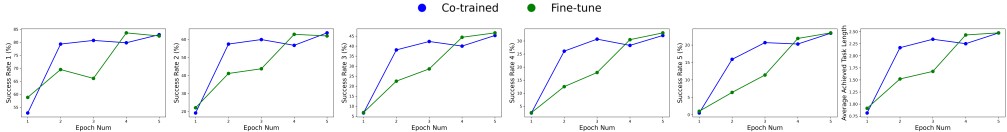

Figure 5: The performance of Co-Trained and Fine-tune model of MPT-3B-IFT at each epoch on $ABC \rightarrow D$ split.

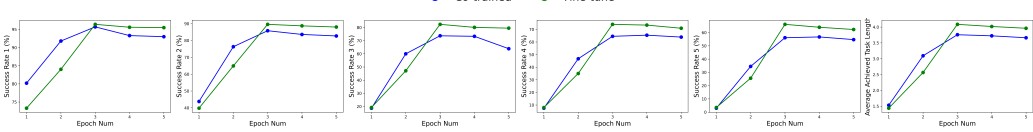

Figure 6: The performance of Co-Trained and Fine-tune model of MPT-3B-IFT at each epoch on $ABCD \rightarrow D$ split.

### B.2 COMPARISON WITH PRE-TRAINED ROBOTICS REPRESENTATION MODELS

We consider comparing our *RoboFlamingo* with recent pre-trained robotics representation models, such as R3M (Nair et al., 2022) and Voltron (Karamcheti et al., 2023). We loaded the pre-train

Table 5: Comparison of co-trained models and fine-tuned models on the CALVIN benchmark. All results are selected from the best of the last 5 epochs.

| Method | Training Data | Test Split | Task Completed in a Sequence | | | | | Avg Len |
|---|---|---|---|---|---|---|---|---|
| | | | 1 | 2 | 3 | 4 | 5 | |
| Co-trained | ABC | D | **0.829** | **0.636** | 0.453 | 0.321 | 0.234 | 2.47 |
| Fine-tune | ABC | D | 0.824 | 0.619 | **0.466** | **0.331** | **0.235** | **2.48** |
| Co-trained | ABCD | D | 0.957 | 0.858 | 0.737 | 0.645 | 0.561 | 3.76 |
| Fine-tune | ABCD | D | **0.964** | **0.896** | **0.824** | **0.740** | **0.66** | **4.09** |
| Co-trained | ABCD | D (Enriched) | 0.678 | 0.452 | 0.294 | 0.189 | 0.117 | 1.73 |
| Fine-tune | ABCD | D (Enriched) | **0.720** | **0.480** | **0.299** | **0.211** | **0.144** | **1.85** |

Table 6: Comparison of co-trained models and fine-tuned models on the COCO image caption and VQAv2 evaluation dataset. All results are selected from the epoch as in Tab. 5.

| Method | COCO | | | | | | | | VQA |
|---|---|---|---|---|---|---|---|---|---|
| | BLEU-1 | BLEU-2 | BLEU-3 | BLEU-4 | METEOR | ROUGE_L | CIDEr | SPICE | Acc |
| Fine-tune (3B, zero-shot) | 0.157 | 0.052 | 0.018 | 0.008 | 0.038 | 0.147 | 0.005 | 0.006 | 4.09 |
| Fine-tune (3B, 4-shot) | 0.168 | 0.057 | 0.020 | 0.008 | 0.043 | 0.161 | 0.005 | 0.007 | 3.87 |
| OpenFlamingo (3B, zero-shot) | 0.580 | 0.426 | 0.301 | 0.209 | 0.208 | 0.464 | 0.757 | 0.153 | 40.92 |
| OpenFlamingo (3B, 4-shot) | 0.612 | 0.461 | 0.332 | 0.234 | 0.220 | 0.491 | 0.822 | 0.162 | 43.86 |
| Co-Train (3B, zero-shot) | 0.223 | 0.157 | 0.106 | 0.071 | 0.124 | 0.334 | 0.346 | 0.084 | 36.37 |
| Co-Train (3B, 4-shot) | 0.284 | 0.204 | 0.142 | 0.098 | 0.142 | 0.364 | 0.426 | 0.100 | 38.73 |
| Original Flamingo (80B, fine-tuned) | - | - | - | - | - | - | 1.381 | - | 82.0 |

weights of R3M and Voltron and fine-tuned them on CALVIN data, only training the policy head while freezing their representation parameters. As for Voltron, we also include a version that fine-tunes the representation layers. The results are shown in Tab. 7, which reveals the clear advantage of fine-tuning pre-trained VLMs compared with these specific robotics representation models.

Table 7: Comparative performance of various VL representation models on various settings, all results are selected from the best of the last 5 epochs.

| Method | Training Data | Test Split | Task Completed in a Sequence | | | | | Avg Len |
|---|---|---|---|---|---|---|---|---|
| | | | 1 | 2 | 3 | 4 | 5 | |
| Voltron (Frozen) | ABC | D | 0.026 | 0.001 | 0.000 | 0.000 | 0.000 | 0.03 |
| Voltron (Fine-tuned) | ABC | D | 0.569 | 0.272 | 0.105 | 0.038 | 0.014 | 1.00 |
| *RoboFlamingo* (Ours) | ABC | D | **0.824** | **0.619** | **0.466** | **0.331** | **0.235** | **2.48** |
| R3M (Frozen) | ABCD | D | 0.085 | 0.005 | 0.001 | 0.000 | 0.000 | 0.10 |
| Voltron (Frozen) | ABCD | D | 0.101 | 0.003 | 0.001 | 0.000 | 0.000 | 0.11 |
| Voltron (Fine-tuned) | ABCD | D | 0.837 | 0.566 | 0.352 | 0.208 | 0.115 | 2.08 |
| *RoboFlamingo* (Ours) | ABCD | D | **0.964** | **0.896** | **0.824** | **0.740** | **0.662** | **4.09** |

### B.3 FINE-TUNE THE FULL MODEL

In the fine-tuning of *RoboFlamingo*, we follow the training of Flamingo (Alayrac et al., 2022; Awadalla et al., 2023) that only trains the parameters of the resampler, the gated cross-attention module of each decoder layer, and the policy head while freezing all other parameters. This leads RoboFlamingo to have 1B trainable parameters (as shown in Tab. 2). In this part, we show the results of training the full model (the MPT-3B-IFT variant), which has 3B trainable parameters in Tab. 8, revealing an obvious performance deterioration.

### B.4 PERFORMANCE CURVES IN TRAINING OF DIFFERENT BACKBONES

Fig. 8 and Fig. 9 show the performance of *RoboFlamingo* with different VLMs on both $ABC \rightarrow D$ and $ABCD \rightarrow D$ settings in 5 training epochs. It is noticed that most variants converge in 5-

Table 8: Comparison between full-model fine-tuning (3B trainable parameters) and *RoboFlamingo*-style fine-tuning (1B trainable parameters)

| Method | Training Data | Test Split | Task Completed in a Sequence (Success Rate) | | | | | |
|---|---|---|---|---|---|---|---|---|
| | | | 1 | 2 | 3 | 4 | 5 | Avg Len |
| Full model fine-tuned | ABCD (Lang) | D | 0.415 | 0.070 | 0.009 | 0.002 | 0.001 | 0.50 |
| *RoboFlamingo* | | | **0.964** | **0.896** | **0.824** | **0.740** | **0.66** | **4.09** |

epoch training and achieve the best performance, benefiting from the pre-training on extensive vision-language tasks.

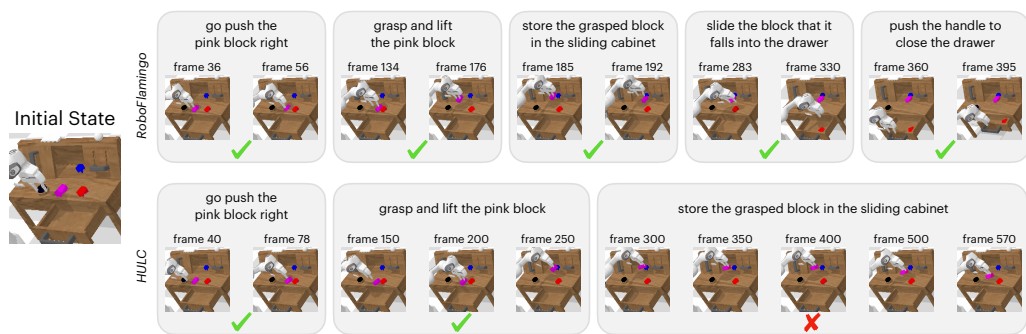

Figure 7: The visualization of *RoboFlamingo* and HULC excuting the same task sequence in the $ABC \rightarrow D$ split.

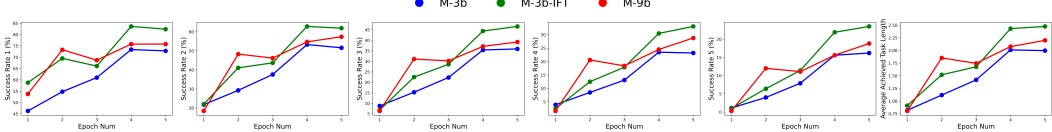

Figure 8: The performance of VLMs at each epoch on $ABC \rightarrow D$ split.

### B.5 QUALITATIVE EXAMPLES

We visualize the task frames and analyze how *RoboFlamingo* achieve such a great performance. As the example shown in Fig. 7, where *RoboFlamingo* successfully finishes the entire task sequence, while HULC stucks at the third one. *RoboFlamingo* only takes a dozen steps to locate and move to the top of the drawer, and simultaneously releases the gripper to complete the task; while HULC keeps moving above the desktop for hundreds of steps and fails to locate the drawer. Furthermore, although both methods are successful for the first two tasks, *RoboFlamingo* uses significantly fewer steps. This representative episode vividly illustrates that our method is much more effective and efficient and could better generalize to unseen vision context.

### B.6 DETAILED IMITATION PERFORMANCES ON EACH TASK

We present the detailed imitation performances by tasks in Tab. 9. All model are reported by their best checkpoint.

### B.7 ROLLOUT EXAMPLES

We present some rollout examples of *RoboFlamingo* on the $ABCD \rightarrow D$ split.

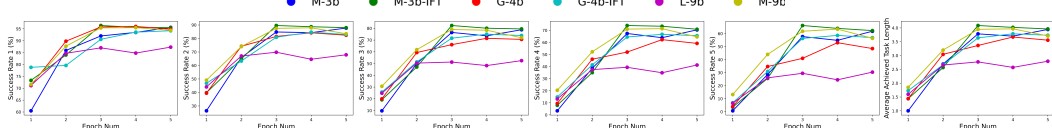

Figure 9: The performance of VLMs at each epoch on $ABCD \rightarrow D$ split.

Table 9: Success rates by task of variants of *RoboFlamingo*. Each task is evaluated 100 times.

| Task Name | M-3B | M-3B-IFT | G-4B | G-4B-IFT | L-9B | M-9B |
|---|---|---|---|---|---|---|
| rotate blue block right | 0.947 | 0.893 | 0.729 | 0.770 | 0.493 | 0.882 |
| move slider right | 0.996 | 0.993 | 0.996 | 0.992 | 0.987 | 0.996 |
| lift red block slider | 0.890 | 0.970 | 0.967 | 0.858 | 0.856 | 0.927 |
| place in slider | 0.904 | 0.828 | 0.582 | 0.911 | 0.874 | 0.910 |
| turn off lightbulb | 0.972 | 1.000 | 0.992 | 0.956 | 0.927 | 0.964 |
| turn off led | 0.988 | 1.000 | 1.000 | 0.994 | 0.970 | 0.981 |
| push into drawer | 0.777 | 0.821 | 0.731 | 0.770 | 0.705 | 0.703 |
| lift blue block drawer | 1.000 | 0.950 | 1.000 | 1.000 | 0.917 | 0.737 |
| close drawer | 1.000 | 1.000 | 1.000 | 1.000 | 0.986 | 0.995 |
| lift pink block slider | 0.940 | 0.971 | 0.944 | 0.862 | 0.861 | 0.918 |
| lift pink block table | 0.859 | 0.851 | 0.905 | 0.892 | 0.543 | 0.899 |
| move slider left | 0.996 | 0.996 | 1.000 | 1.000 | 0.970 | 0.996 |
| open drawer | 0.976 | 0.997 | 0.997 | 0.982 | 0.980 | 0.997 |
| turn on lightbulb | 0.988 | 0.994 | 1.000 | 0.988 | 0.949 | 0.988 |
| rotate blue block left | 0.923 | 0.939 | 0.820 | 0.925 | 0.636 | 0.848 |
| push blue block left | 0.746 | 0.955 | 0.841 | 0.836 | 0.677 | 0.909 |
| rotate red block right | 0.926 | 0.972 | 0.853 | 0.905 | 0.591 | 0.959 |
| turn on led | 0.988 | 0.988 | 0.994 | 0.976 | 0.985 | 0.994 |
| push pink block right | 0.652 | 0.754 | 0.833 | 0.651 | 0.627 | 0.750 |
| push red block left | 0.949 | 0.920 | 0.849 | 0.949 | 0.562 | 0.908 |
| lift blue block table | 0.891 | 0.956 | 0.925 | 0.927 | 0.611 | 0.931 |
| place in drawer | 0.988 | 0.989 | 0.988 | 0.975 | 0.971 | 0.976 |
| rotate red block left | 0.970 | 0.908 | 0.950 | 0.953 | 0.677 | 0.952 |
| push pink block left | 0.947 | 0.920 | 0.915 | 0.973 | 0.747 | 0.933 |
| stack block | 0.612 | 0.641 | 0.608 | 0.595 | 0.569 | 0.604 |
| lift blue block slider | 0.847 | 0.963 | 0.908 | 0.826 | 0.769 | 0.869 |
| push red block right | 0.657 | 0.732 | 0.797 | 0.451 | 0.457 | 0.653 |
| lift red block table | 0.948 | 0.939 | 0.942 | 0.975 | 0.606 | 0.989 |
| lift pink block drawer | 0.857 | 0.800 | 0.929 | 0.714 | 0.778 | 0.923 |
| rotate pink block right | 0.917 | 0.896 | 0.714 | 0.794 | 0.478 | 0.789 |
| unstack block | 1.000 | 0.982 | 0.957 | 0.980 | 0.946 | 0.979 |
| rotate pink block left | 0.929 | 0.839 | 0.906 | 0.818 | 0.698 | 0.927 |
| push blue block right | 0.479 | 0.597 | 0.471 | 0.478 | 0.400 | 0.594 |
| lift red block drawer | 0.947 | 1.000 | 1.000 | 1.000 | 0.769 | 0.933 |

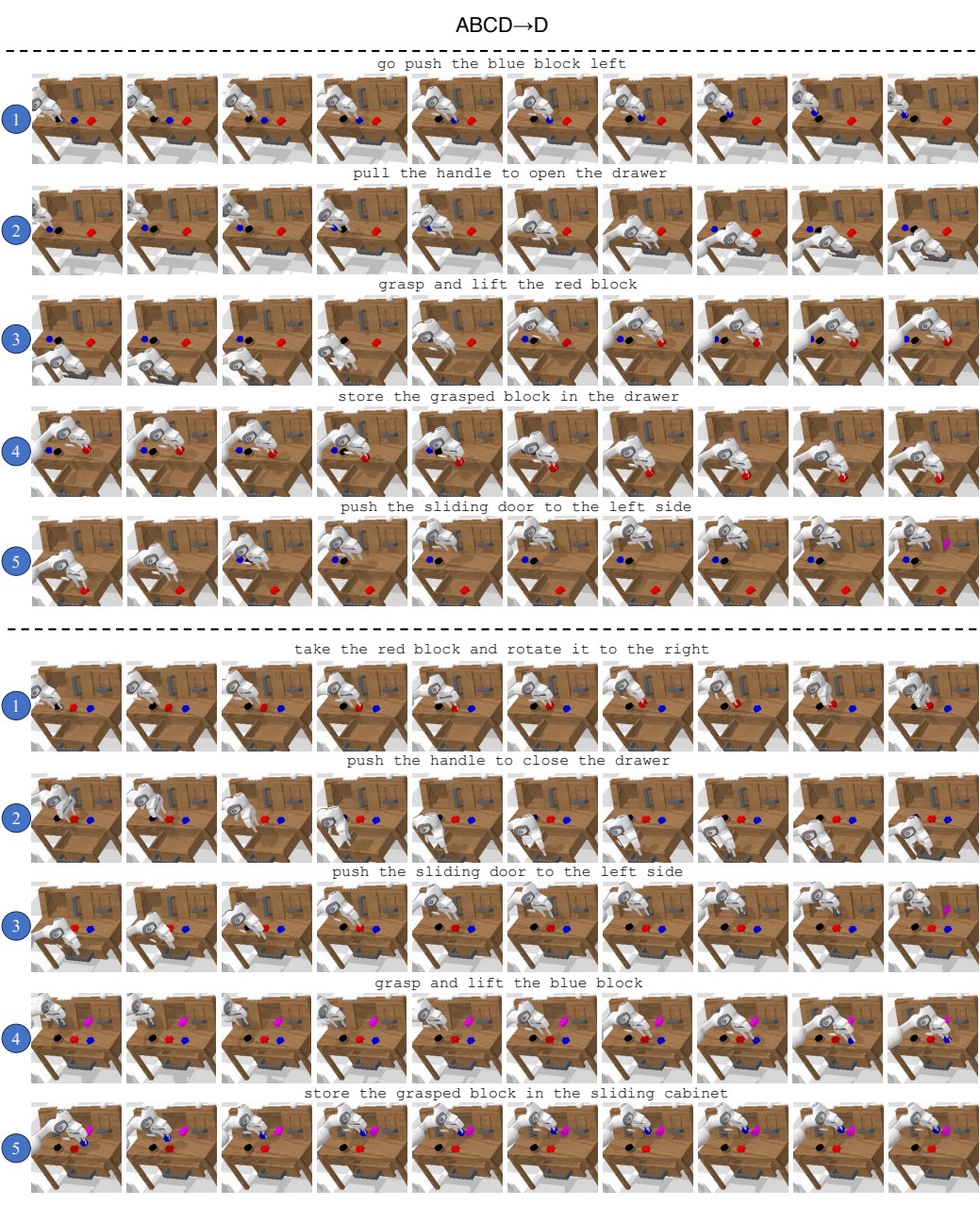

Figure 10: Rollouts on the $ABCD \rightarrow D$ split of the CALVIN benchmark.

# C ADDITIONAL DETAILS

## C.1 ILLUSTRATION OF POLICY HEADS/FORMULATION

We illustrate the details of the four policy heads/formulation mentioned in Section 5.4: (a) $MLP\ w/o\ hist$ takes only the current observation as input to predict actions, which ignores the observation history. (b) $MLP\ w\ hist$ takes the history frames into the vision encoder with position embedding, and encodes the history information through the cross-attention layers in the feature fusion decoder. (c) $GPT$ and (d) $LSTM$ both utilize the VLM backbone to process single-frame observations and integrate the history with the policy head. $GPT$ explicitly takes the visual history as input to predict the next action. $LSTM$ implicitly maintains a hidden state to encode memory and predict the action.

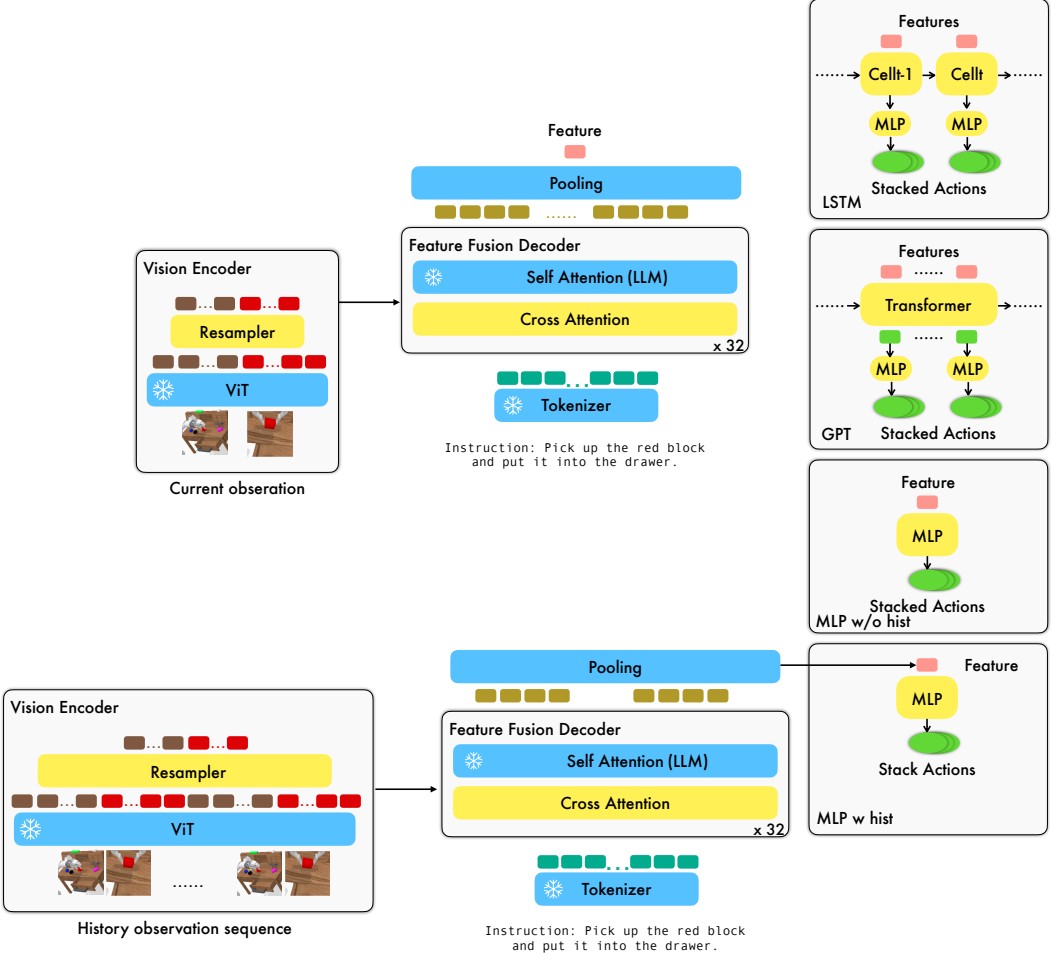

Figure 11: Implementation details of all policy heads/formulation involved.

