# OpenReview forum: "Vision-Language Foundation Models as Effective Robot Imitators"
_ICLR.cc/2024/Conference — ICLR 2024 spotlight_

### Official Review · Reviewer_iuWF · 2023-10-29

**Soundness:** 3 good
**Presentation:** 3 good
**Contribution:** 3 good
**Rating:** 6
**Confidence:** 4

**Summary:**

The paper uses a VLM as a backbone for visual imitation learning of language-conditioned policies. In contrast to prior work, it builds on open-source VLMs and limits the number of finetuned parameters to make VLM-policy training feasible for conventional compute budgets. They demonstrate that the VLM backbone leads to superior imitation performance on the CALVIN simulated robotic manipulation benchmark and compare a few different design choices.

**Strengths:**

Using VLM backbones for policy learning is a promising direction for robot learning, but prior work was confined to proprietary models and required large compute for policy training. This work's focus on open-source models and parameter-efficient finetuning brings those models within the reach of academic compute budgets and thus is very valuable. The demonstrated results on the CALVIN benchmark are strong and support the claim that pre-trained VLM backbones are good features for imitation learning.

I appreciate that the authors analyzed several of the design decisions experimentally and showed which choices have a larger influence on final performance. Particularly the results in Table 3 are interesting in that they show that larger VLM backbones are particularly beneficial in a low-data regime. I also appreciated the separate investigation of generalizability in the visual domain and to diverse language instructions.

The paper is easy to follow and most of the experiments are easily understandable.

**Weaknesses:**

A main selling point of the paper is that it claims the introduced method can forego expensive co-finetuning by restricting the number of finetuned weights and freezing most of the VLM weights. However, if I understand correctly, the paper finetunes all weights that were also finetuned in the Flamingo VLM grounding stage, i.e. like Flamingo they froze the vision and language model features, but finetuned all cross-attention features that perform the vision-to-language grounding (and Fig 3b shows that this is crucial). This however suggest that the model may still forget most of the knowledge obtained in the VLM pretraining stage, ie the OpenFlamingo training. The experimental section of the paper lacks comparison to (A) co-training with the current parameter-freezing scheme, (B) full model finetuning w/ and w/o co-training to support the claim that their partial finetuning scheme is actually key to enable good performance without co-finetuning.

Another comparison that would be good to add is to a simpler, pre-trained visual representation, like VC-1, Voltron etc. These models also use internet data to train good representations for imitation learning, but are arguably easier to use than the billion-parameter scale models introduced here, so it would be good to experimentally show the benefits.

One notable difference to prior work is that instead of predicting actions as tokens in the VLM's output vocabulary, the proposed method trains a separate action head. It would be good to analyze this choice and compare to directly outputting actions as "text tokens".

Since the paper is mainly an empirical study, it would be good to evaluate the policy on more than one environment, e.g. the IKEA Furniture assembly environment could be a nice and challenging testbed with ~ photorealistic rendering.

The paper also lacks details on the computational requirements for training policies with the VLM backbones (required GPU + training time), which seems crucial given the focus on making VLM policies more accessible.

Finally, Section 5.5 on open-loop execution lacks some detail on what exactly was tried, so I was a bit confused about these experimental results (see questions below).

**Questions:**

- for the enriched language evaluations, the authors mention that they sample language instruction synonyms randomly from the GPT-4 generations -- did you ensure that all methods are evaluated on the same randomly sampled set of instructions to make the comparison fair?

- can you explain in more detail the experiment on open-loop execution in Section 5.5? How can you open-loop execute the policy without re-training?


## Review Summary

Overall I think this paper is an interesting contribution to democratizing the access to large vision language models for policy learning. I believe that many in the community will be interested in this and thus recommend acceptance. However, the empirical analysis in the paper could be significantly improved by addressing the points raised above. Concretely, the authors can:
- add details about the required compute
- add comparison to co-finetuning and full model finetuning
- add comparison to other pre-trained representations
- add evaluations on at least one additional environment

**Details Of Ethics Concerns:**

--

# Post Rebuttal Comments

Thanks for answering my review!

I appreciate the new experiments. For the full finetuning experiment -- is it possible that the model is overfitting to the small target dataset? It could be interesting to show the validation curve over time and potentially perform early stopping?

For the IKEA furniture env: note that there is a new version of the environment -- even though it's focused on real world execution, it also comes with a simulated version of the environment and offline data for imitation I believe (https://clvrai.github.io/furniture-bench/)

The compute requirements are a bit disheartening to see -- a 8xA100 (80GB) server should be able to perform full finetuning for models in the 7-13B scale with proper FSDP. For making these models truly accessible, pushing the compute requirements down to ~1x48GB GPU would be ideal so I'd encourage the authors to further push to reduce the requirements.

I will maintain my score and recommend acceptance due to the relevance of the research to the community.

---

> ### Author Response · Authors · 2023-11-21
> **Thank you very much for your affirmative review of our work and very helpful suggestions!**
>
> Following your advice, we have updated:
> 1. **Added computational requirements for training policies** in Appendix A.3. "All experiments involved in this paper are conducted on a single GPU server with 8 NVIDIA Tesla A100 GPUs, and the default batch size is 6 on each GPU. The MPT-3b model takes 13 hours of training per epoch and achieves the best performance at the 3rd epoch, while the MPT-9b model also takes 26 hours of training per epoch and achieves the best performance at the 3rd epoch."
> 2. **Added the comparison results with pre-trained robotics representation work Voltron and R3M**. In our implementation, we freeze the representation layers of R3M and Voltron. We also tried to fine-tune the entire model of Voltron to improve its performance. Please see the results in the table for comparison and Appendix B.2 of our paper for more details and further analysis:
> |  | Split | SR 1 | SR 2 | SR 3 | SR 4 | SR 5 | Avg Len |
> | --- | --- | --- | --- | --- | --- | --- | --- |
> | Voltron  | ABC->D | 2.6% | 0.1% | 0.0% | 0.0% | 0.0% | 0.027 |
> | Voltron (Unfreeze) | ABC->D | 56.9% | 27.2% | 10.5% | 3.8% | 1.4% | 0.998 |
> | Ours | ABC->D | 82.4% | 61.9% | 46.6% | 33.1% | 23.5% | 2.475 |
> | R3M | ABCD->D | 8.5% | 0.5% | 0.1% | 0.0% | 0.0% | 0.098 |
> | Voltron | ABCD->D | 10.1% | 0.3% | 0.1% | 0.0% | 0.0% | 0.105 |
> | Voltron (Unfreeze) | ABCD->D | 83.7% | 56.6% | 35.2% | 20.8% | 11.5% | 2.078 |
> | Ours | ABCD->D | 96.4% | 89.6% | 82.4% | 74.0% | 66.2% | 4.086 |
>
> 3. **Added full model finetuning to the 3B version** (cannot do bigger size model for now) on the $ABCD\rightarrow D$ setting. The current version of RoboFlamingo trained 1B parameters (as shown in Table 2) while keeping all others frozen on a single 8 A100 GPU server, which takes a batch size of 6 on each 80G A100 GPU (data parallel). Therefore, we temporarily have not extended to the 9B variant due to the development of the model parallel. The results are listed below, which indicate that the performance will deteriorate so much if we do so.
> |  | SR 1 | SR 2 | SR 3 | SR 4 | SR 5 | Avg Len |
> | --- | --- | --- | --- | --- | --- | --- |
> | Full model fine-tuned (3B) | 41.5% | 7.0% | 0.9% | 0.2% | 0.1% | 0.497 |
>
>
> 4. **Added the co-train verion model with both robot data and vision-language data (COCO caption, VQA)**. The results shown below indicate that co-training could preserve most ability of the VLM backbone on vl tasks, while losing a bit of performance on robot tasks. These results have been updated in Appendix B.1.
> |  | Split | SR 1 | SR 2 | SR 3 | SR 4 | SR 5 | Avg Len |
> | --- | --- | --- | --- | --- | --- | --- | --- |
> | Co-Train | ABC->D | 82.9% | 63.6% | 45.3% | 32.1% | 23.4% | 2.473 |
> | Fine-tune | ABC->D | 82.4% | 61.9% | 46.6% | 33.1% | 23.5% | 2.475 |
> | Co-Train | ABCD->D | 95.7% | 85.8% | 73.7% | 64.5% | 56.1% | 3.758 |
> | Fine-tune | ABCD->D | 96.4% | 89.6% | 82.4% | 74.0% | 66.2% | 4.086 |
> | Co-Train | ABCD->D (Enrich) | 67.8% | 45.2% | 29.4% | 18.9% | 11.7% | 1.73 |
> | Fine-tune | ABCD->D (Enrich) | 72.0% | 48.0% | 29.9% | 21.1% | 14.4% | 1.854 |
>
> |  | coco  |  |  |  |  |  |  |  | VQA |
> | --- | --- | --- | --- | --- | --- | --- | --- | --- | --- |
> |  | BLEU-1 | BLEU-2 | BLEU-3 | BLEU-4 | METEOR | ROUGE_L | CIDEr | SPICE | Acc |
> | Fine-tune (3B, zero-shot) | 0.157  | 0.052 | 0.018 | 0.008 | 0.038 | 0.147 | 0.005 | 0.006 | 4.09  |
> | Fine-tune (3B, 4-shot) | 0.168  | 0.057 | 0.020 | 0.008 | 0.043 | 0.161 | 0.005 | 0.007 | 3.87 |
> | OpenFlamingo (3B, zero-shot)  | 0.580 | 0.426 | 0.301  |  0.209 |  0.208  |  0.464  |  0.757  |  0.153  |  40.92
> | OpenFlamingo (3B, 4-shot)  | 0.612 | 0.461 | 0.332 | 0.234 | 0.220 | 0.491 | 0.822 | 0.162 | 43.86 |
> | Co-Train (3B, zero-shot) | 0.223 | 0.157 | 0.106 | 0.071 | 0.124 | 0.334 | 0.346 | 0.084 | 36.37 |
> | Co-Train (3B, 4-shot) | 0.284 | 0.204 | 0.142 | 0.098 | 0.142 | 0.364 | 0.426 | 0.100 | 38.73 |
> | Original Flamingo (80B, fine-tuned) | - | - | - | - | - | - | 1.381 | - | 82.0 |

---

> ### Author Response · Authors · 2023-11-21
> **Response cont.**
>
> Here we answer your questions below.
>
> **Q1**: `...It would be good to analyze this choice and compare to directly outputting actions as "text tokens".`
>
> **A1**: We highly agree with you, and we indeed have thought about doing such experiments. However, since the major difference between RT-2 and RoboFlamingo comes from the foundation model side, we find it hard to do so under the architecture of Flamingo, which is simply a VL model, not a VLA model (the Palm-E used in RT-2). Therefore, if we cannot use the VLM to model history action tokens from the input side, but only predict discrete action tokens instead, the difference will only come from regression loss or discrete token classification loss, which can not specify any advantage of utilizing action tokens or not.
> In fact, in our initial submitted version, the MLP w/ hist version is similar to the implementation that directly outputs text tokens. However, as stated, this version still uses LSTM to encode history observations. We further implement a version that is closer to RT-2 by discretizing each dim of the action into 256 bins and decoding the action for each DoF. However, we still can not get a reasonable performance for this model (the performance for the single-task is lower than 10%, and the Avg. Len. Is less than 0.1). We are still trying our best to make it work.
>
> **Q2**: `for the enriched language evaluations, the authors mention that they sample language instruction synonyms randomly from the GPT-4 generations -- did you ensure that all methods are evaluated on the same randomly sampled set of instructions to make the comparison fair?`
>
> **A2**: Yes absolutely, we generate a new file containing a sequence of instructions, and all methods are evaluated over those regenerated instructions.
>
> **Q3**: `can you explain in more detail the experiment on open-loop execution in Section 5.5? How can you open-loop execute the policy without re-training?`
>
> **A3**: Sorry for your confusion! The policy head is trained to predict action sequences (stacked actions) instead of a single-step action. In our closed-loop control, only the next action will be executed and then we give the new observation to the policy head to predict the following actions. In contrast, for open-loop control, we execute an action sequence before providing a new observation to the policy head after a few timesteps. We've revised the related context and Figure 2 to make it more clear.
> "Instead of taking only the next action to execute and performing VLM inference every time for new observations to predict future actions, open-loop control can be achieved by predicting an action sequence (stacked actions) with only one inference given the current observation, therefore alleviating the delay and the test-time computing requirement."
>
> **Q4**: `The IKEA dataset.`
>
> **A4**: We sincerely appreciate your valuable suggestion. We acknowledge that extending our model's validation to additional benchmarks, such as the photo-realistic IKEA dataset, would indeed strengthen our findings. However, the IKEA dataset, primarily an online resource lacking linguistic annotations, presents significant challenges in accumulating sufficient offline data for effective model training. Presently, constraints in time and computational resources preclude us from undertaking this extensive data collection process. Consequently, we consider this an important avenue for future research and intend to explore it as soon as the necessary resources are available.

---

### Official Review · Reviewer_fijq · 2023-10-31

**Soundness:** 3 good
**Presentation:** 2 fair
**Contribution:** 3 good
**Rating:** 6
**Confidence:** 4

**Summary:**

This paper proposes the RoboFlamingo architecture for effective language-conditioned robot manipulation task learning through behavior cloning. Specifically, the paper shows that by initiating model weights from pretrained VLMs and finetuning them in the OpenFlamingo-style using a minimal amount of downstream robot manipulation data, the policy can achieve good performance on the CALVIN benchmark for both seen task and unseen task variations. Such performance also outperforms previous baselines like RT-1. The authors further provide ablations on the effect of different backbone scales, different architectures, and training paradigms on agent performance.

**Strengths:**

- The authors agree to open-source their code and implementations, which I really appreciate. This will greatly facilitate efforts to scale up foundation model training for robotic manipulation tasks.
- The ablation section is very helpful for readers to understand the critical components of the proposed RoboFlamingo architecture.
- RoboFlamingo significantly outperforms prior baselines like RT-1 on the CALVIN benchmark.

**Weaknesses:**

Firstly, the presentation of the paper can be improved, and some parts of the method is unclear, which hinders reader's understanding.
- According to Section 4.2.2, the finetuned language model backbone will output a fused vision-language representation $X_t^L=\{x_{t,1}^L,\dots, x_{t,M}^L\}$ for *each* time step $t \in [1...T]$, where $M$ is the length of the input language instruction $l$. To produce such outputs, it seems necessary that the same language instruction needs to be tiled $T$ times and fed into the language model for a sequence of $T$ images. However, such detail is not illustrated in Figure 2, and Figure 2 only illustrates the model behavior when $T=1$. It would be a lot more helpful if Figure 2 illustrates model behavior when $T>1$.
- Sec. 5.4.2 of the ablation study shows that "loading the pre-trained parameters of the cross-attention layers" is crucial for model performance. From which model are these cross-attention layer weights being loaded? Also why not load the perceiver resampler weights from a pretrained model? In addition, the design to load pretrained cross-attention weights is never described in the methodology section.
- In Section 5.4.3, what is the setup for the "instruction fine-tuning" experiment? I also didn't find the descriptions for the instruction finetuning designs in the methodology section.

Secondly, authors choose to only train RoboFlamingo on the CALVIN benchmark throughout the paper. Even though authors claim that they want to showcase RoboFlamingo's ability to produce good language-conditioned manipulation policies given a small amount of finetune data, I'm afraid that by only finetuning on the CALVIN benchmark, the model overfits to the dataset and loses some crucial abilities like spatial reasoning and object relation understanding (which may be crucial for other robot manipulation tasks that are not present in the CALVIN benchmark). By training RoboFlamingo on a mixture of downstream robot datasets and large-scale datasets used to pretrain e.g., OpenFlamingo, InstructBLIP, IDEFICS, LLaVA, authors might alleviate such phenomenon, and even improve upon the CALVIN benchmark performance they achieved in this paper. Therefore, I do not quite agree with the author's claim that "only a minimal amount of data is required to adapt the model to downstream manipulation tasks".

**Questions:**

Page 3: `Compared to other works, the controlling policies do not require any ability to understand instructions, but rely on the pre-trained frozen LLM to select necessary skills.`. This sentence is inaccurate. The author's proposed approach still need to understand language instructions as the LLM backbone still needs to fuse language representations with visual input representations. I believe the authors' actual meaning is that the policy head does not explicitly take language instructions as input.

---

> ### Author Response · Authors · 2023-11-21
> **Thanks for your detailed check and suggestions!**
>
> Thanks for your detailed check and suggestions! We've fixed all the mentioned typos and notations, and keep revising the manuscript. For your concerns:
>
> **Q1**: `To produce such outputs, it seems necessary that the same language instruction needs to be tiled...It would be a lot more helpful if Figure 2 illustrates model behavior when $T>1$`
>
> **A1**:
> 1) The VLM backbone only models single-step observation and the same language instruction is modeled in every single step.
> 2) Thanks for your suggestion! We've added the caption for Fig. 2 **"The Flamingo backbone models single-step observations, and the temporal features are modeled by the policy head."**, and added a figure in Appendix C.1, Fig.9 to illustrate the policy head in detail.
>
> **Q2**: `Sec. 5.4.2 of the ablation study shows that "loading the pre-trained parameters of the cross-attention layers" is crucial for model performance. From which model are these cross-attention layer weights being loaded? Also why not load the perceiver resampler weights from a pretrained model? In addition, the design to load pretrained cross-attention weights is never described in the methodology section.`
>
> **A2**: Sorry for making you confused. First, we apologize that the most accurate description should be **"without loading the pre-trained parameters of the cross-attention layers and the resampler trained by OpenFlamingo models"**. This means we only use a pre-trained ViT encoder and a pre-trained self-attention layer provided by those LLMs, and train everything else from scratch.
> Basically, this is to validate if the VL pretraining is useful. The architecture of Flamingo (and its open-sourced version OpenFlamingo), as shown in Fig. 2, utilizes a pre-trained ViT encoder (from CLIP) and a pre-trained LLM (with only self-attention layers). So both the resampler and the cross-attention layers are the extra parameters provided by the pretrained OpenFlamingo VLMs. In our work, we follow the fine-tuning of Flamingo by 1) loading the pre-trained parameters **from OpenFlamingo VLMs**; and 2) only training the resampler and the cross-attention layer (as mentioned in the last paragraph of section 4.4).
> We have changed Figure 2 and supplement related descriptions at the beginning of section 4.2 to make this more clear. Thanks for your comments!
>
> **Q3**: `In Section 5.4.3, what is the setup for the "instruction fine-tuning" experiment? I also didn't find the descriptions for the instruction finetuning designs in the methodology section.`
>
> **A3**: Sorry for your confusion! In this subsection we study the critical factors in VL pre-training (OpenFlamingo) instead of the fine-tuning of RoboFlamingo. For clarity, we change the context to the following:
> Instruction fine-tuning is a specialized technique to help LLMs perform specific tasks according to explicit instructions. We find that LLMs with such a training stage can improve the performance of the policy in both seen and unseen scenarios, revealed by the performance improvements of M-3b-IFT against M-3b, and G-4b-IFT against G-4b shown in Table 2.

---

> ### Author Response · Authors · 2023-11-21
> **Response cont.**
>
> **Q4**:  `I'm afraid that by only finetuning on the CALVIN benchmark, the model overfits to the dataset and loses some crucial abilities like spatial reasoning and object relation understanding...By training RoboFlamingo on a mixture of downstream robot datasets and large-scale datasets used to pretrain...might alleviate such phenomenon, and even improve upon the CALVIN benchmark performance...Therefore, I do not quite agree with the author's claim that "only a minimal amount of data is required to adapt the model to downstream manipulation tasks".`
>
> **A4**:
> Thanks for your kind review and enlightening suggestions. We agree that joint training with VL datasets may help to alleviate catastrophic forgetting. However, we would like to say that learning with limited manipulative demonstrations is also important since it is expensive to collect large amounts of data on robotic platforms.
> In this paper, we focused more on the second part and found that adapting large-scale VL models to robotic tasks is not too "expensive" in the aspect of collecting robotic demonstrations. But surely, we would be excited to take the first part as the ongoing future work.
> We have revised the claim so that it is more clear and more rigorous: "With such a decomposition, we only need to combine a small amount of robotics demonstration to adapt the model to downstream manipulation tasks". We also conduct experiments on co-training RoboFlamingo with existing open-source visual-language datasets, including coco caption and visual question answering.
> Furthermore, as you ask, we implement a co-training framework for both VL data (COCO image caption, VQA) and robotics data (CALVIN), and the performance on both VL tasks and robot tasks is reported as follows:
> |  | Split | SR 1 | SR 2 | SR 3 | SR 4 | SR 5 | Avg Len |
> | --- | --- | --- | --- | --- | --- | --- | --- |
> | Co-Train | ABC->D | 82.9% | 63.6% | 45.3% | 32.1% | 23.4% | 2.473 |
> | Fine-tune | ABC->D | 82.4% | 61.9% | 46.6% | 33.1% | 23.5% | 2.475 |
> | Co-Train | ABCD->D | 95.7% | 85.8% | 73.7% | 64.5% | 56.1% | 3.758 |
> | Fine-tune | ABCD->D | 96.4% | 89.6% | 82.4% | 74.0% | 66.2% | 4.086 |
> | Co-Train | ABCD->D (Enrich) | 67.8% | 45.2% | 29.4% | 18.9% | 11.7% | 1.73 |
> | Fine-tune | ABCD->D (Enrich) | 72.0% | 48.0% | 29.9% | 21.1% | 14.4% | 1.854 |
>
> |  | coco  |  |  |  |  |  |  |  | VQA |
> | --- | --- | --- | --- | --- | --- | --- | --- | --- | --- |
> |  | BLEU-1 | BLEU-2 | BLEU-3 | BLEU-4 | METEOR | ROUGE_L | CIDEr | SPICE | Acc |
> | Fine-tune (3B, zero-shot) | 0.157  | 0.052 | 0.018 | 0.008 | 0.038 | 0.147 | 0.005 | 0.006 | 4.09  |
> | Fine-tune (3B, 4-shot) | 0.168  | 0.057 | 0.020 | 0.008 | 0.043 | 0.161 | 0.005 | 0.007 | 3.87 |
> | OpenFlamingo (3B, zero-shot)  | 0.580 | 0.426 | 0.301  |  0.209 |  0.208  |  0.464  |  0.757  |  0.153  |  40.92
> | OpenFlamingo (3B, 4-shot)  | 0.612 | 0.461 | 0.332 | 0.234 | 0.220 | 0.491 | 0.822 | 0.162 | 43.86 |
> | Co-Train (3B, zero-shot) | 0.223 | 0.157 | 0.106 | 0.071 | 0.124 | 0.334 | 0.346 | 0.084 | 36.37 |
> | Co-Train (3B, 4-shot) | 0.284 | 0.204 | 0.142 | 0.098 | 0.142 | 0.364 | 0.426 | 0.100 | 38.73 |
> | Original Flamingo (80B, fine-tuned) | - | - | - | - | - | - | 1.381 | - | 82.0 |
>
> From the result, we observe that co-training could indeed help the model preserve its original ability over VL tasks. This provides a solution for fine-tuning VLMs to robotics models while preserving the ability on vision-language tasks, even though it may slightly deteriorate the performance on robotic tasks.
>
> **Q5**:  `Page 3: Compared to other works, the controlling policies do not require any ability to understand instructions, but rely on the pre-trained frozen LLM to select necessary skills. This sentence is inaccurate...I believe the authors' actual meaning is that the policy head does not explicitly take language instructions as input.`
>
> **A5**: We think you may misunderstand our statement. **This part talks about the related works (not our work)**, which utilize LLMs to plan the language goals into detailed, pre-defined low-level goals. Those low-level goals will be handled by a pre-trained skill policy, which does not need to understand the instructions (because LLM handles that part).

---

> ### Comment · Reviewer_fijq · 2023-11-22
>
> Thanks authors for the rebuttal. The figures and the network initialization are now clearer (Fig. 11 in Appendix C1 is especially helpful).
>
> For the instruction finetuning ablation in Sec. 5.4.3, I believe misunderstandings comes from the term of "instruction finetuning" appearing in this section for the first time in the paper, which might make readers think that besides training RoboFlamingo on the CALVIN dataset, there is another separate instruction finetuning process, rather than the fact that training RoboFlamingo on the CALVIN dataset is already the instruction finetuning process. Thus, I would suggest authors to state earlier in the paper that training RoboFlamingo on the CALVIN dataset (which is essentially, finetuning the pretrained Flamingo model and training an additional temporal policy head using CALVIN instruction-observation-action triplets), will be referred to as "instruction finetuning" later on.
>
> For the additional experiments where authors co-train RoboFlamingo on COCO and VQAv2, authors observe a drop in CALVIN benchmark performance after co-training. How is the training data balanced in a single batch? Also I feel using e.g., LLaVA 1.5 mixture of instruction tuning data for co-training might yield better results, but due to the tight rebuttal deadline I wouldn't expect authors to finish the experiment by then.
>
> However, I'm willing to raise my final rating to 6.

---

> > ### Author Response · Authors · 2023-11-23
> > **Thank you for your ongoing engagement with our research. We value this opportunity to clarify aspects of our experiment, specifically regarding the Instruction-Finetuning and CoTraining settings.**
> >
> > We regret any confusion caused by our initial description of Instruction-Finetuning. This term is intended to describe a pre-training phase in Large Language Models. To illustrate, both MPT-3b and MPT-3b-IFT employ the same network architecture as the VLM backbone. The distinction lies in the pre-training data: MPT-3b’s LLM is pre-trained on the Redpajama Dataset, whereas MPT-3b-IFT’s LLM undergoes additional finetuning with the databricks-dolly-15k dataset. This open-source dataset comprises instruction-following records created by Databricks employees, covering various behavioral categories like brainstorming, classification, and summarization, among others. Based on the pretrained language models, OpenFlamingo further adapts to vision-language tasks.
> >
> > To summarize, our training strategies for both models are exactly the same. The only difference is that we used different initial weights. We have revised the text in section 5.4.3 from `Instruction fine-tuning is a specialized technique to help LLMs perform specific tasks according to explicit instructions.` to `Instruction-Finetuning is a specialized technique that utilizes a further pre-training enhancement on the LLM with the IFT dataset, which provides a rich repertoire of instruction-following behaviors that inform its capabilities in language-conditioned tasks.` to enhance clarity on this point.
> >
> > Regarding the Co-Train setting detailed in our paper, we ensured that the model equally incorporates batches of VL and robot data in each epoch. This approach is now more thoroughly explained in Appendix B.1. We concur with your suggestion that leveraging Vision-Language instruction finetuning datasets, such as LLaVa 1.5, could potentially improve our results, given its extensive data variety and task diversity. We are committed to updating our experimental findings with results from co-training on LLaVA 1.5 as soon as the process is complete.

---

### Official Review · Reviewer_qVoh · 2023-11-01

**Soundness:** 4 excellent
**Presentation:** 3 good
**Contribution:** 3 good
**Rating:** 6
**Confidence:** 4

**Summary:**

This work introduces RoboFlamingo, a language-conditioned manipulation method that finetunes an open-source VLM OpenFlamingo to output low-level robot control actions. The method builds upon the pre-trained and frozen OpenFlamingo Transformer backbone: 1) adds an LSTM policy head after the pooled visual-language embedding output from the OpenFlamingo backbone, 2) adds first-person and third-person camera image tokens to the ViT for the Resampler. Following the OpenFlamingo finetuning procedure, the ViT, Tokenizer, and Self-Attention Layers of the backbone are frozen during training; only the resampler, cross-attention, and policy head parameters are updated during finetuning on robot imitation learning datasets. In evaluations, RoboFlamingo is evaluated on: 1) in-distribution training performance and out-of-distribution generalization on the CALVIN benchmark where it achieves SOTA over HULC and RT-1, 2) ablations that show history based policy heads (GPT and LSTM) outperform MLPs, vision-language pretraining is critical for good performance, and 3) larger models and instruction finetuned base models performing better. The authors commit to releasing code upon acceptance.

**Strengths:**

- The motivation is clear for a low-cost alternative solution to large closed Vision Language Action models (VLAs) like RT-2, which motivate this work. The study which incorporates open-source design components like the different LLMs of various architectures and sizes in OpenFlamingo is a great contribution to the open-sourced community as well.
- The results on CALVIN, a well established and difficult robot control and generalization benchmark, are very compelling
- Open loop results are intriguing for pragmatic on-robot deployment
- The presentation is largely very easy to follow and a pleasure to read

**Weaknesses:**

- Other ways of incorporating VL pre-training are not considered, such as utilizing VL representations like R3M or VOLTRON or MVP. These baselines are relevant given the frozen-backbone + robot finetuning setup in RoboFlamingo. Essentially, a baseline should study different ways of incorporating "web data", which the current baselines do not study.
- A core claim of RT-2 was the benefit of co-fine-tuning on robotics data in addition to the original VL data. This core claim is not studied in RoboFlamingo.
- Another claim of RT-2 was measuring the transfer of internet knowledge to robotics, in addition to in-domain performance. This seems like a major benefit of utilizing VLMs for robotics generalization. However, this is not studied in this work; the setting in the ABC => D CALVIN environment seems insufficient to measure how much transfer is occuring from internet-scale VL pre-training to robotics.
- Another claim of RT-2 was the benefit of mixing robot action tokens explicitly with VL tokens. In contrast, RoboFlamingo introduces a new policy head that directly only predicts action tokens. It would be interesting to compare an explicit action-only policy head with the multi-modal output token prediction setting in RT-2.
- The presentation can be improved a bit in Section 4.2.1 and 4.2.2, where the notation is unwieldy. For example, the notation of $K$ is overloaded.
- Writing nits:
    - Section 2: "models to encoder" => "models to encode", "train the policy" => "training the policy", "utilizing robot manipulation data both the web data" => "utilizing both robot manipulation data and web data", "We hope RoboFlamingo provide" => "We hope RoboFlamingo provides"
    - Section 4: "Particularlly" => "Particularly", "look into one" => "looks into one", "and take its" => "and takes its"
    - Section 5: "We wonder" => "We study", "24 thousand trajectories" => weird ~ added
    - Section 5.4.1: "single-frame observation" => "single-frame observations"

**Questions:**

- Clarifications to my concerns above would be appreciated.
- Will checkpoints be released as well?
- How does performance on pre-training tasks change during finetuning? That is, is there catastrophic forgetting occurring, where the base foundation capabilities are lost?

---

> ### Author Response · Authors · 2023-11-21
> **Thanks for your detailed review! All your mentioned typos and abused notations are now fixed and we are trying our best to revise the manuscript.**
>
> ## Response to Questions
>
> **Q1**: `Will checkpoints be released as well?`
>
> **A1**: Certainly! The whole project is trying our best to contribute to the open-source community.
>
> **Q2**: `How does performance on pre-training tasks change during finetuning? That is, is there catastrophic forgetting occurring, where the base foundation capabilities are lost?`
>
> **A2**: We appreciate your insightful reviews. We've revealed some evidence that the model lost some foundation capabilities, as shown in Table 1, Enriched Lang setting (the last four rows), where we use GPT-4 to generate new instructions with the same meaning. The performance loss indicates that there is over-fitting during fine-tuning. To further understand the phenomenon, we conduct further experiments by testing the fine-tuned RoboFlamingo model (the M-3b-IFT variant) on the COCO and VQAv2, which verifies our conjecture that it does overfit the robotics dataset and loses many original VL abilities.
> To prevent such problems, we choose to co-train RoboFlamingo (the M-3b-IFT variant) with VQA and COCO datasets during fine-tuning of the robotics dataset. We test the co-train model on both CALVIN and the COCO and VQAv2 tasks as well. This provides a solution for fine-tuning VLMs to robotics models while preserving the ability on vision-language tasks, even though it may slightly deteriorate the performance on robotic tasks. One interesting observation is that under the Enriched Lang setting, the performance of the co-trained model also drops. This may indicate the difference between understanding different sentences and aligning vision-language representations (as the pre-trained tasks do).
> See the following tables for the detailed numerical results.
>
> **Results on CALVIN**
> |  | Split | SR 1 | SR 2 | SR 3 | SR 4 | SR 5 | Avg Len |
> | --- | --- | --- | --- | --- | --- | --- | --- |
> | Co-Train | ABC->D | 82.9% | 63.6% | 45.3% | 32.1% | 23.4% | 2.473 |
> | Fine-tune | ABC->D | 82.4% | 61.9% | 46.6% | 33.1% | 23.5% | 2.475 |
> | Co-Train | ABCD->D | 95.7% | 85.8% | 73.7% | 64.5% | 56.1% | 3.758 |
> | Fine-tune | ABCD->D | 96.4% | 89.6% | 82.4% | 74.0% | 66.2% | 4.086 |
> | Co-Train | ABCD->D (Enrich) | 67.8% | 45.2%  | 29.4% | 18.9% | 11.7% | 1.73 |
> | Fine-tune | ABCD->D (Enrich) | 72.0% | 48.0% | 29.9% | 21.1% | 14.4% | 1.854 |
>
>
> |  | coco  |  |  |  |  |  |  |  | VQA |
> | --- | --- | --- | --- | --- | --- | --- | --- | --- | --- |
> |  | BLEU-1 | BLEU-2 | BLEU-3 | BLEU-4 | METEOR | ROUGE_L | CIDEr | SPICE | Acc |
> | Fine-tune (3B, zero-shot) | 0.157  | 0.052 | 0.018 | 0.008 | 0.038 | 0.147 | 0.005 | 0.006 | 4.09  |
> | Fine-tune (3B, 4-shot) | 0.168  | 0.057 | 0.020 | 0.008 | 0.043 | 0.161 | 0.005 | 0.007 | 3.87 |
> | OpenFlamingo (3B, zero-shot)  | 0.580 | 0.426 | 0.301  |  0.209 |  0.208  |  0.464  |  0.757  |  0.153  |  40.92
> | OpenFlamingo (3B, 4-shot)  | 0.612 | 0.461 | 0.332 | 0.234 | 0.220 | 0.491 | 0.822 | 0.162 | 43.86 |
> | Co-Train (3B, zero-shot) | 0.223 | 0.157 | 0.106 | 0.071 | 0.124 | 0.334 | 0.346 | 0.084 | 36.37 |
> | Co-Train (3B, 4-shot) | 0.284 | 0.204 | 0.142 | 0.098 | 0.142 | 0.364 | 0.426 | 0.100 | 38.73 |
> | Original Flamingo (80B, fine-tuned) | - | - | - | - | - | - | 1.381 | - | 82.0 |
>
> ## Response to Weakness
>
> **Q1**: `Other ways of incorporating VL pre-training are not considered, such as utilizing VL representations like R3M or VOLTRON or MVP. These baselines are relevant given the frozen-backbone + robot finetuning setup in RoboFlamingo.`
>
> **A1**: Thanks for your great advice! We have supplemented the experiment results of R3M and Voltron on CALVIN benchmark, which are both pre-trained robotics representation models.
> In our implementation, we freeze the vision and text encoder of R3M and Voltron. We also fine-tune the entire model of Voltron to improve its performance. Please see the results in the table for comparison:
>
> |  | Split | SR 1 | SR 2 | SR 3 | SR 4 | SR 5 | Avg Len |
> | --- | --- | --- | --- | --- | --- | --- | --- |
> | Voltron  | ABC->D | 2.6% | 0.1% | 0.0% | 0.0% | 0.0% | 0.027 |
> | Voltron (Unfreeze) | ABC->D | 56.9% | 27.2% | 10.5% | 3.8% | 1.4% | 0.998 |
> | Ours | ABC->D | 82.4% | 61.9% | 46.6% | 33.1% | 23.5% | 2.475 |
> | R3M | ABCD->D | 8.5% | 0.5% | 0.1% | 0.0% | 0.0% | 0.098 |
> | Voltron | ABCD->D | 10.1% | 0.3% | 0.1% | 0.0% | 0.0% | 0.105 |
> | Voltron (Unfreeze) | ABCD->D | 83.7% | 56.6% | 35.2% | 20.8% | 11.5% | 2.078 |
> | Ours | ABCD->D | 96.4% | 89.6% | 82.4% | 74.0% | 66.2% | 4.086 |

---

> ### Author Response · Authors · 2023-11-21
> **Response cont.**
>
> **Q2**: `A core claim of RT-2 was the benefit of co-fine-tuning on robotics data in addition to the original VL data. This core claim is not studied in RoboFlamingo.` `Another claim of RT-2 was measuring the transfer of internet knowledge to robotics, in addition to in-domain performance. This seems like a major benefit of utilizing VLMs for robotics generalization. However, this is not studied in this work...`
>
> **A2**: Thanks for your reminder! The main point and contribution of this paper is to show a way of fine-tuning open-source VLMs to robotics models. We believe RT2's conclusions should also benefit RoboFlamingo. In our implementation, we co-train our RoboFlamingo model with both robot data and vision-language data (COCO image caption, VQA). From the results, we observe that co-training could preserve most ability of the VLM backbone on VL tasks, while losing some performance on robot tasks. We have the results performed in the following table and updated in Appendix B.1 of our paper.
> |  | Split | Epoch | SR 1 | SR 2 | SR 3 | SR 4 | SR 5 | Avg Len |
> | --- | --- | --- | --- | --- | --- | --- | --- | --- |
> | CoTrain | ABC->D | 1 | 52.8% | 19.2% | 6.6% | 2.4% | 0.5% | 0.815 |
> |  | ABC->D | 2 | 79.3% | 57.4% | 38.1% | 26.1% | 15.9% | 2.168 |
> |  | ABC->D | 3 | 80.7% | 59.9% | 42.3% | 30.7% | 20.7% | 2.343 |
> |  | ABC->D | 4 | 79.8% | 56.7% | 40.0% | 28.3% | 20.3% | 2.251 |
> |  | ABC->D | 5 | 82.9% | 63.6% | 45.3% | 32.1% | 23.4% | 2.473 |
> | Finetune | ABC->D | 5 | 82.4% | 61.9% | 46.6% | 33.1% | 23.5% | 2.475 |
> | CoTrain | ABCD->D | 1 | 80.2% | 43.8% | 18.7% | 7.5% | 3.0% | 1.532 |
> |  | ABCD->D | 2 | 91.8% | 76.3% | 60.0% | 46.7% | 34.5% | 3.093 |
> |  | ABCD->D | 3 | 95.7% | 85.8% | 73.7% | 64.5% | 56.1% | 3.758 |
> |  | ABCD->D | 4 | 93.3% | 83.6% | 73.3% | 65.4% | 56.7% | 3.723 |
> |  | ABCD->D | 5 | 93.0% | 82.7% | 63.9% | 63.9% | 54.7% | 3.662 |
> | Finetune | ABCD->D | 3 | 96.4% | 89.6% | 82.4% | 74.0% | 66.2% | 4.086 |
>
> **Q3**: `It would be interesting to compare an explicit action-only policy head with the multi-modal output token prediction setting in RT-2.`
>
> **A3**: We highly agree with you, and we indeed have thought about doing such experiments. However, since the major difference between RT-2 and RoboFlamingo comes from the foundation model side, we find it is hard to do so under the architecture of Flamingo, which is simply a VL model, not a VLA model (the Palm-E used in RT-2). Therefore, if we cannot use the VLM to model history action tokens from the input side, but only predict discrete action tokens instead, the difference will only come from regression loss or discrete token classification loss, which can not specify any advantage of utilizing action tokens or not.
>
> For your concern, we have attempted to apply token mixing configurations like RT-2. However, this approach did not converge when using token cross-entropy loss. We suspect this issue stems from the architectural differences in the VLM backbone between RoboFlamingo and RT-2. Given that RT-2 is not publicly accessible, reproducing its results is beyond our current scope due to time constraints.

---

> > ### Comment · Reviewer_qVoh · 2023-11-22
> >
> > Thanks for your response! You have addressed many of my questions. I am more positive about this work than previously.
> >
> > I maintain my rating and still remain in favor of acceptance. I may further adjust my rating during discussions with other reviewers.

---

### Official Review · Reviewer_8RDp · 2023-11-02

**Soundness:** 3 good
**Presentation:** 3 good
**Contribution:** 3 good
**Rating:** 8
**Confidence:** 4

**Summary:**

- Proposes RoboFlamingo, a framework for adapting large vision-language models (VLMs) like OpenFlamingo to robot manipulation policies.
- Achieves state-of-the-art performance on the CALVIN benchmark by fine-tuning VLMs with only a small amount of robotic demonstration data.
- Shows VLMs can enable effective vision-language comprehension and long-horizon planning for robot control when combined with a simple policy head, while demonstrating strong generalization ability to unseen tasks and environments. Comprehensive analysis and ablation studies on using VLMs for robotic manipulation are conducted.

**Strengths:**

- RoboFlamingo outperforms considerably prior methods on CALVIN
- Requires much less data and compute than methods like RT-2 that co-train on extensive internet-scal data.
- Decouples perception and policy to enable flexibility like open-loop control, while maintaining relatively strong zero-shot generalization ability.

**Weaknesses:**

- Relies on simulated robot environment, may be challenging to transfer to real world.
- The evaluation is limited to a single simulated benchmark environment (CALVIN). Testing on more diverse robotic platforms and tasks in simulation could help validate the generalizability of the method.
- Less sample efficient than methods leveraging offline robot data like MCIL.

**Questions:**

- What steps would be needed to transfer RoboFlamingo to real robotic systems? How realistic are the CALVIN simulations?
- Is there scope to incorporate offline robotic data to improve sample efficiency?
- Experiments are with visual and language modalities. Robotic manipulation often relies on additional sensing (e.g. force, tactile). How can RoboFlamingo incorporate other modalities?
- How flexible is the decoupled design? Will it be possible to incorporating RoboFlamingo into hierarchical frameworks like in PaLM-E?
- How does the computational overhead of RoboFlamingo compare to other VLM-based methods?

---

> ### Author Response · Authors · 2023-11-21
> **Thanks for your affirmative review of this paper! We hereby answer your questions as follows**
>
> **Q1**: `Relies on simulated robot environment`, `How realistic are the CALVIN simulations?` `limited to a single simulated benchmark environment (CALVIN)`
>
> **A1**: We admit the submitted version of the manuscript limits in the simulation benchmark CALVIN. The CALVIN tasks are implemented based on Pybullet (https://pybullet.org/), which is based on the Bullet Physics SDK and has been used in many previous robotics works (e.g., [1-3]). This benchmark provides diversified robotic manipulation tasks including rotate blocks, turn off lights, open drawers, stack and unstack blocks, and so on, which are still challenging to current state-of-the-art algorithms. Here are some intuitive demonstrations (http://calvin.cs.uni-freiburg.de/images/banner.mp4). We hope that the additional materials will be helpful in understanding the reality, fidelity, and challenges of this benchmark.
>
> **Q2**: `What steps would be needed to transfer RoboFlamingo to real robotic systems?`
>
> **A2**: We tried our best to train the RoboFlamingo on real-world robotics tasks. To do so, we collect real-robot demonstrations for table-top manipulation tasks including pick-and-place, opening drawer and cabinet, etc. As a result, we first collected around 5% CALVIN-size demonstrations using teleoperation and training to finetune RoboFlamingo as we did on the CALVIN benchmark. The robot can hardly manipulate objects successfully due to severe overfitting. To make it work, we then collect an additional set of 5% CALVIN-size demonstrations, based on which we have successfully trained a model that can manipulate some objects in pick-and-place tasks. Nevertheless, we found that this dataset is still not big enough to train a high-performance model at such a scale. We are still working on that, including data collection and model training. However, due to the limited timeline of the review process, we can not provide comprehensive results here. But from performance improvements from 5% CALVIN-size data to 10%CALVIN-size data, we believe that our model can adapt to real-world applications.
>
> **Q3**: `Is there scope to incorporate offline robotic data to improve sample efficiency?`
>
> **A3**: According to your enlightening suggestions, we have verified that more offline robotic data will further improve performance, with less training epochs the performance is improved.
> To get more data, we used the script provided by the CALVIN benchmark to generate 4x more data (5x in total), then we trained RoboFlamingo on that, and observed that there is a clear performance gain compared to the model trained on the standard CALVIN dataset. For MPT-3B-IFT, the performance increases to 4.253 on the ABCD -> D setting in the aspect of Avg. Len, with a success rate of 97.1% for the single-task setting and 71.6% for the subsequent 5-task setting. For MPT-9B, the performance increases to 4.095 on the ABCD -> D setting in the aspect of Avg. Len, with a success rate of 95.4% for the single-task setting and 69.4% for the subsequent 5-task setting.
> Due to the time limit, we only evaluate the models' performance for 2 epochs, and we believe that the performance would keep increasing through incorporating more offline robotic data and training more iterations.
> This table reports the raw performance and the performance of the same model trained on 5x data:
>
> | Model | Epoch | SR 1 | SR 2 | SR 3 | SR 4 | SR 5 | Avg Len |
> | --- | --- | --- | --- | --- | --- | --- | --- |
> | MPT-3b-IFT | best in the paper (3 epoch) | 96.4% | 89.6% | 82.4% | 74.0% | 66.2% | 4.086 |
> |  | 1 | 96.1% | 87.5% | 80.1% | 72.1% | 65.4% | 4.012 |
> |  | 2 | 97.1% | 91.6% | 85.6% | 79.4% | 71.6% | 4.253 |
> | MPT-9b | best in the paper (4 epoch) | 95.5% | 87.9% | 78.4% | 71.4% | 63.4% | 3.966 |
> |  | 1 | 95.4% | 88.0% | 81.4% | 75.3% | 69.4% | 4.095 |
> |  | 2 | 95.1% | 88.6% | 80.5% | 73.9% | 66.9% | 4.05 |
>
> **Q4**: `Experiments are with visual and language modalities. Robotic manipulation often relies on additional sensing (e.g. force, tactile). How can RoboFlamingo incorporate other modalities?`
>
> **A4**: Very nice and interesting point! Currently, our RoboFlamingo is only evaluated on gripper-based manipulation tasks, which do not need to involve force and tactile information. Since such modalities are orthogonal to visual and language modalities, modeling such data with pre-trained VLMs may require much more effort to fine-tune, so we think the most straightforward and simple way is to treat it as low-dimensional states, concatenated with the visual-language features in the policy head input.

---

> ### Author Response · Authors · 2023-11-21
> **Response cont.**
>
> **Q5**: `How flexible is the decoupled design? Will it be possible to incorporating RoboFlamingo into hierarchical frameworks like in PaLM-E?`
>
> **A5**: As mentioned in the abstract and introduction, the decoupled design makes full use of the ability of pre-trained VLMs to understand single-step visual and language data, and allows for open-loop control and deployment on low-performance platforms (big VLMs inferring on the cloud and policy head runs at the device side). This is very interesting and has great potential to be further investigated.
>
> **Q6**: `How does the computational overhead of RoboFlamingo compare to other VLM-based methods?`
>
> **A6**: All experiments involved in this paper are conducted on a single GPU server with 8 NVIDIA Tesla A100 GPUs. The training data includes ~2.4M interaction steps, and the trainable parameters for all RoboFlamingo variants are only 1M in total. The MPT-3b model takes 13 hours of training per epoch and achieves the best performance at the 3rd epoch, while the MPT-9b model also takes 26 hours of training per epoch and achieves the best performance at the 3rd epoch. For comparison, RT2 uses the PaLI-X 5B & 55B model, PaLI 3B model and the PaLM-E 12B model, and is deployed on a multi-TPU cloud server.
> We have supplemented this information in the updated revised version, Appendix A.3.
>
> **Q7**: `Less sample efficient than methods leveraging offline robot data like MCIL.`
>
> **A7**: Since we are utilizing the VLM foundation models, the amount of trainable parameters is much larger than the baseline models like MCIL.  However, by incorporating additional training techniques including adding adapters, freezing language embedding, predicting multiple-step actions, and predicting end-effector relative actions, we find that we can improve the performance of RoboFlamingo on the limited robotics data (10% training data, Table 3 in the paper) setting and achieve better performance than HULC. The results are listed as follows:
> |  | SR 1 | SR 2 | SR 3 | SR 4 | SR 5 | Avg Len |
> | --- | --- | --- | --- | --- | --- | --- |
> | RT-1 | 12.3% | 1.5% | 0.1% | 0.0% | 0.0% | 0.139 |
> | HULC | 70.0% | 28.5% | 9.4% | 3.3% | 0.7% | 1.119 |
> | MPT-9b | 54.7% | 19.0% | 6.7% | 2% | 0.3% | 0.83 |
> | MPT-9b + additional training techniques | 68.7% | 30.1%  | 10.2% | 2.8% | 1.3% | 1.131 |

---

### Author Response · Authors · 2023-11-21
**Overall response**

We thank all reviewers for their detailed reviews and suggestions! All mentioned typos and abused notations are fixed and we've tried our best to revise the whole manuscript. A new version of the pdf is updated.

Overall, in the replaced submission, we have updated:

1. **Added computational requirements for training policies** in Appendix A.3. "All experiments involved in this paper are conducted on a single GPU server with 8 NVIDIA Tesla A100 GPUs, and the default batch size is 6 on each GPU. The MPT-3b model takes 13 hours of training per epoch and achieves the best performance at the 3rd epoch, while the MPT-9b model also takes 26 hours of training per epoch and achieves the best performance at the 3rd epoch."

2. **Added the comparison results with pre-trained robotics representation work Voltron and R3M** in Appendix B.2.

3. **Added full model fine-tuning to the 3B version** in Appendix B.3.

4. **Added co-finetuning experiments by training RoboFlamingo on both robot data and vision-language data (COCO caption, VQA)** in Appendix B.1.

5. **Added a detailed policy structure illustration** in Appendix C.1.

---

### Meta-Review · Area_Chair_BtWX · 2023-12-11

**Metareview:**

The paper proposes a novel (and simple) framework on robot manipulation using vision-language models. The work shows strong performance, and all reviewers appreciate the contribution. During the rebuttal phrase, the authors provided more evidence and addressed most concerns raised. Based on the reviews and discussions, the area chair recommends accepting the submission.

**Justification For Why Not Higher Score:**

The proposed method aligns with many concurrent works to use vision-language model for long-horizon manipulation tasks.

**Justification For Why Not Lower Score:**

The method is valid and the results have significantly outperformed the past.

---

### Decision · Program_Chairs · 2024-01-16

Accept (spotlight)